# Self-Guidance: Enhancing Neural Codecs via Decoder Manifold Alignment

Xiang Li [1 2]   Yixuan Zhou [1]   Jingran Xie [1 2]   Zhiyong Wu [1]   Hui Wang [2]

## Abstract

Neural speech codecs based on Vector-Quantized VAEs (VQ-VAEs) are core audio tokenizers for speech LLMs, yet their reconstruction fidelity is bottlenecked by quantization error. Modifying the quantizer or increasing model capacity are common fixes, but they complicate downstream language modeling. Our core idea is to align the decoder's internal feature manifolds when processing both the quantized tokens and their original continuous embeddings, using a lightweight feature-mapping loss. This requires minimal training overhead and no inference-time changes. Applied to XCodec2, self-guidance improves all reconstruction metrics, achieving state-of-the-art low-bitrate performance. Notably, it enables a 4× codebook reduction without fidelity loss, which downstream TTS experiments show significantly improves LLM-based synthesis by simplifying the token modeling space. Multiple statistical observations and visualizations corroborate the enhanced internal manifold alignment in the decoder. Extensive experiments confirm its generality across various inductive biases. Self-guidance thus establishes an efficient, broadly applicable method for high-fidelity neural audio coding.

## 1. Introduction

Audio codecs serve as essential tools for audio compression, originally designed to encode continuous audio signals like human speech into sequences of reconstructable discrete codes, enabling efficient data transmission and storage (Wu et al., 2024a). Recently, neural speech codecs, pioneered by SoundStream (Zeghidour et al., 2021) and EnCodec (Défossez et al., 2022), leverage the Vector-Quantized Variational AutoEncoder (VQ-VAE) (Van Den Oord et al., 2017;

Esser et al., 2021) architectures to achieve high-fidelity reconstruction at compression ratios significantly exceeding traditional codecs. This breakthrough facilitates the integration of large language models (LLMs) in speech processing and generation, where the discretized audio tokens could be directly adopted in the standard next-token-prediction frameworks of LLMs. Benefiting from large-scale speech modeling with LLMs, numerous studies have advanced downstream tasks, including text-to-speech generation (Wang et al., 2023; Yang et al., 2023b) and interactive multimodal large language models (MLLMs) (Défossez et al., 2024; Zhan et al., 2024).

The transformation from continuous audio to discrete tokens in a VQ-VAE is enabled by a latent vector quantizer. This component maps continuous latent vectors from the encoder to entries in a finite codebook via nearest-neighbor search (i.e., vector quantization) (Van Den Oord et al., 2017; Yu et al., 2021; Mentzer et al., 2023). The corresponding codebook embeddings are then passed to the decoder to reconstruct the audio waveform.

However, quantization is inherently lossy. As noted in prior work (Liu et al., 2024) and confirmed by our preliminary experiments (Section 3.2), the decoder produces higher-fidelity audio when using the continuous, pre-quantized latents compared to the quantized tokens. This performance gap confirms that quantization error constitutes a major obstacle to high-fidelity reconstruction, as it restricts the information available to the decoder.

To suppress quantization error, existing neural codecs typically employ strategies such as hierarchical quantization with multiple residual codebooks (Zeghidour et al., 2021; Yang et al., 2023a) or simply scaling up the codebook size (Parker et al., 2024; Xin et al., 2024; Wu et al., 2024b; Ye et al., 2025a). While effective for compression, these approaches introduce significant challenges for downstream LLM modeling. Hierarchical codebooks require complex mechanisms to fit within auto-regressive transformer frameworks (Wang et al., 2023; Yang et al., 2023b; Défossez et al., 2024). On the other hand, unlike text tokenizers that use hierarchical subword units (e.g., BPE) (Dubey et al., 2024), *a larger audio codebook expands a flat, unstructured vocabulary*, exponentially increasing the complexity of autoregressive sequence modeling (Ye et al., 2025b).

[1]Shenzhen International Graduate School, Tsinghua University, Shenzhen, China [2]Pengcheng Laboratory, Shenzhen, China. Correspondence to: Zhiyong Wu <zywu@sz.tsinghua.edu.cn>.

*Proceedings of the 43rd International Conference on Machine Learning*, Seoul, South Korea. PMLR 306, 2026. Copyright 2026 by the author(s).

Consequently, reducing quantization error often necessitates complex codec designs that shift the modeling burden downstream. We introduce a different perspective: instead of solely improving the quantizer, we can **make the decoder robust to its imperfections**. Our method trains the decoder to align its internal representations—and thus its outputs—when processing both the original continuous embeddings and the quantized tokens. By directly bridging this gap, we mitigate quantization artifacts at the decoder, alleviating the pressure on the quantizer itself.

To this end, we propose a novel learning scheme for VQ-VAE-based codecs, which we call **self-guidance (SG)**. During training, the decoder receives both the quantized token embeddings and the continuous pre-quantized latent vectors. We then apply a feature mapping loss between the decoder's intermediate features or outputs for these two paths. This additional objective uses the high-fidelity output from the pre-quantized latents as a target, guiding the decoder to produce similar, high-quality features when driven by the quantized tokens. Consequently, the decoder becomes more robust to quantization artifacts, enhancing the final reconstructed audio's fidelity.

We implement our self-guidance approach on the state-of-the-art single-codebook neural speech codec XCodec2 (Ye et al., 2025b), applying the feature mapping loss to the outputs of the decoder's transformer backbone. Experiments on LibriSpeech show consistent reconstruction improvements across various codebook sizes (8192, 16384, 65536), vector quantizers (FSQ, SimVQ, Residual FSQ), and decoder networks (Transformer-based XCodec2, RNN/CNN-based BigCodec). Notably, we achieve comparable reconstruction quality with only a quarter of the original codebook size. The benefits of a smaller codebook are further demonstrated in downstream text-to-speech LLM experiments. Audio samples are available on our demo website.[1]

Our main contributions are as follows:

1. We propose a self-guidance mechanism for the VQ-VAE backbone of neural codecs that aligns the decoder's internal manifolds to counteract quantization error, and provide statistical evidence confirming this alignment effect rather than encoder regulation.

2. Applying self-guidance to XCodec2 achieves state-of-the-art reconstruction for low-bitrate speech codecs, with extensive experiments demonstrating its generalization across various inductive biases.

3. We show that self-guidance reduces dependency on large codebooks, enabling higher compression rates and significant benefits for downstream LLM-based applications.

---

[1] https://sgvqvae.github.io/sgvqvae-demo

## 2. Related Works

### 2.1. Vector Quantization

VQ-VAE (Van Den Oord et al., 2017) introduced discrete latent representations for generative models, and VQ-VAE2 (Razavi et al., 2019) enhanced representation richness through hierarchical architectures. VQGAN (Esser et al., 2021) integrated adversarial networks, establishing a fundamental VQ framework for high-quality generative models such as Stable Diffusion (Rombach et al., 2022). Nevertheless, these methods encounter representation collapse when dealing with large codebook sizes, which restricts their scalability.

To tackle this issue, DALL-E (Ramesh et al., 2021) employs Gumbel-Softmax sampling to activate more codes during training, although only a small subset of codes is used for quantization during inference (Zhang et al., 2023). VQGAN-FC (Yu et al., 2021) mitigates collapse by reducing latent dimensionality and applying L2 normalization. Finite scaler quantization (FSQ) (Mentzer et al., 2023) and its variant Look-up free quantization (LFQ) (Yu et al., 2023) project latents to low-dimensional spaces (e.g., binary codes), but this comes at the cost of model capacity, as performance degrades when codebooks are small or collapse is not severe. Recently, VQGAN-LC (Zhu et al., 2024a) and SimVQ (Zhu et al., 2024b) enable stable training with codebook sizes up to 100k by incorporating a linear projector for the codebook.

### 2.2. Neural Codec

In early neural codec model studies, SoundStream (Zeghidour et al., 2021) utilized residual vector quantizers (RVQs) to distribute the codec model's total bitrate across multiple codebooks, preventing codebook size explosion. However, this hierarchical design complicates downstream applications due to the multiple tokens within each frame, necessitating additional flattening or joint modeling.

In recent years, single-codebook codecs have emerged as a simpler and more efficient alternative, demonstrating strong performance at low bitrates (Li et al., 2024; Guo et al., 2024; Ji et al., 2024; Xin et al., 2024; Della Libera et al., 2025). For instance, BigCodec (Xin et al., 2024) employs larger model sizes and advanced learning objectives to achieve high-fidelity audio decoding from a single quantizer of frame rate 80Hz. Despite these advancements, the reconstruction fidelity of BigCodec on perspective metrics significantly degrades at lower frame rates. While high frame rate incurs longer audio token sequences, resulting in a quadratic increase in downstream LLM computation cost, and the language modeling complexity (Wang et al., 2024).

To address this challenge, XCodec (Ye et al., 2025a) and FocalCodec (Della Libera et al., 2025) integrate pretrained self-supervised audio encoders to support the reconstruction

performance on perspective metrics. Thanks to the stabilized quantizer like FSQ, TS3Codec (Wu et al., 2024b) and XCodec2 (Ye et al., 2025b) extend the codebook size to over $2^{16}$ to further boost the model performance, achieving state-of-the-art performance on single-codebook codec of frame rate around 50Hz. However, the drastically extended codebook size poses a significant challenge to the language modeling of downstream LLMs. This issue motivates the development of this paper to explore an approach that relieves the existing codec model's dependency on the large codebook.

### 2.3. Self-distillation

Self-distillation is a specific paradigm within knowledge distillation where the student and teacher are instances of the same model architecture, or even the same model itself (Zhang et al., 2019; Furlanello et al., 2018). This is often achieved by using a model's own outputs from a previous training iteration or a differently initialized copy as the teacher's knowledge, leveraging consistency regularization to improve the student's generalization and calibration (Mobahi et al., 2020). Our method is conceptually related to self-distillation, which also uses feature-mapping losses for model guidance. However, self-guidance presents a distinct contribution by specifically targeting the decoder's robustness to quantization error in VQ-VAEs—a previously under-explored bottleneck. Our core innovation lies in using the pre-quantized features as an internal guide to explicitly align the decoder's manifolds, establishing a new training paradigm for speech codecs. Furthermore, unlike self-distillation, which requires a pre-trained teacher, our approach is implemented within a single, end-to-end training process, offering greater practicality and efficiency.

## 3. Preliminary: Quantization Artifacts

### 3.1. Revisiting the VQ-VAE Framework

The Vector-Quantized Variational Autoencoder (VQ-VAE) forms the foundation of modern neural audio codecs. As illustrated in Figure 1 (left), the architecture consists of three main components: an encoder, a vector quantizer, and a decoder. The encoder processes an input audio signal $x$ to produce a sequence of continuous latent embeddings $z_e \in \mathbb{R}^{d_e}$, where $d_e$ is the latent dimension. The vector quantizer then maps each embedding in $z_e$ to the nearest entry in a finite codebook $\mathcal{Q} \subset \mathbb{R}^{d_e}$. This operation produces a sequence of quantized token embeddings $z_q$. Finally, the decoder reconstructs the audio signal $\hat{x}$ from $z_q$. During training, gradients are propagated through the non-differentiable quantization operation using straight-through estimation (STE), which copies gradients from $z_q$ directly to $z_e$.

The quantization process inherently introduces error as it projects continuous latent vectors onto a discrete codebook. The **quantization error** can be quantified as:

$$e_q = \|z_e - z_q\|_2 \tag{1}$$

This error represents the information loss incurred during discretization.

*Table 1.* Comparing the reconstruction performance of neural speech codec models with different decoder inputs.

| Codec Model | bitrate (kbps) | decoder input | STOI↑ | WER↓ | SIM↑ |
|---|---|---|---|---|---|
| *Ground Truth* | | | 1.00 | 2.4 | 1.000 |
| Encodec | 6 | $z_q$ | 0.88 | 2.7 | 0.861 |
| Encodec | 6 | $z_e$ | **0.95** | 2.7 | **0.922** |
| BigCodec | 1.04 | $z_q$ | 0.93 | 3.6 | 0.841 |
| BigCodec | 1.04 | $z_e$ | **0.95** | **2.9** | **0.872** |

### 3.2. Observation of Quantization Artifacts

The quantization error $e_q$ introduces information loss that propagates to the decoder, resulting in reconstruction fidelity degradation, know as the **quantization artifacts**. This phenomenon is evident even though the decoder is exclusively trained on quantized inputs during standard VQ-VAE training.

As shown in Table 1, according to the findings from Liu et al. (2024) on the EnCodec model (Défossez et al., 2022), when the decoder processes the continuous pre-quantized latents $z_e$ instead of the quantized tokens $z_q$, reconstruction quality improves significantly. This observation aligns with our evaluations of BigCodec (Xin et al., 2024).

These results demonstrate that quantization artifacts substantially limit reconstruction quality, presenting a major obstacle for achieving optimal performance in neural codecs. Thus, it is desirable to enhance the decoder's robustness to the quantization error, enabling the generation of high-fidelity samples from the post-quantized latents despite the quantization error.

## 4. Methodology

To mitigate quantization artifacts in neural speech codecs, we propose a novel learning scheme called **self-guidance** (SG) for VQ-VAE decoders. This section details the self-guidance mechanism and explains our rationale for applying it to the XCodec2 model to construct a high-fidelity neural speech codec.

### 4.1. Self-guidance Mechanism

The self-guidance mechanism is designed to enhance the decoder's ability to compensate for the information loss caused

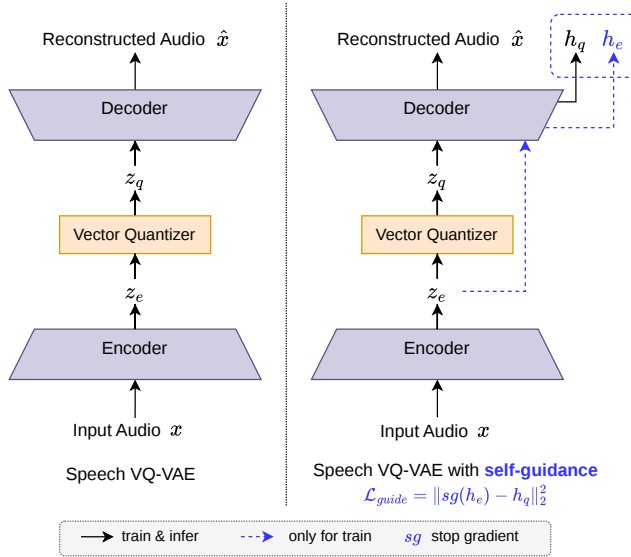

*Figure 1.* Illustration of the VQ-VAE architecture and the proposed self-guidance (SG) mechanism.

by quantization error in the input tokens $z_q$. Specifically, we aim to enable the decoder to produce similar outputs from both the quantized tokens $z_q$ and the continuous pre-quantized latents $z_e$.

While the vanilla VQ-VAE reconstruction loss implicitly guides the decoder toward this objective by using the original input $x$ as a target, our preliminary analysis indicates that this alone is insufficient to fully address quantization artifacts. This suggests the need for more explicit guidance during training.

Inspired by our preliminary findings, we propose using the pre-quantized latent $z_e$ itself as an internal guidance signal. As illustrated in Figure 1 (right), during training we introduce an additional forward pass that feeds $z_e$ to the decoder. We then extract intermediate hidden features from both paths: $h_e$ from the $z_e$ branch and $h_q$ from the $z_q$ branch. Specifically, the feature $h$ is obtained as the output of the final Transformer block in XCodec2 decoder, which is then fed to an ISTFT head to reconstruct waveform. We introduce a feature-mapping loss $\mathcal{L}_{\text{guide}}$ to align these features:

$$\mathcal{L}_{\text{guide}} = \|\text{sg}(h_e) - h_q\|_2^2 \qquad (2)$$

where $\text{sg}(\cdot)$ denotes the stop-gradient operation. This loss term is added to the original VQ-VAE objectives to form an end-to-end self-supervised training process.

The self-guidance mechanism introduces minimal computational and architectural overhead:

- **Training:** Only an additional forward pass through the decoder with $z_e$ is required, with no gradient computa-

tion needed for this branch.

- **Inference:** No modifications are required; the decoder operates exclusively on $z_q$ as in standard VQ-VAE.

### 4.2. Neural Speech Codec Model

To validate the effectiveness of self-guidance, we apply it to XCodec2, a state-of-the-art neural speech codec that has demonstrated strong performance in low-bitrate speech encoding and downstream speech generation tasks (Boson AI, 2025; Ye et al., 2025b).

XCodec2 comprises several key components: a convolutional encoder, a single-layer finite scalar quantizer (FSQ), and an acoustic decoder. Additionally, it includes a semantic encoder and decoder that form an auxiliary autoencoder operating on Wav2Vec2-BERT features (Barrault et al., 2023), enhancing the semantic content of the encoded latents for improved downstream performance.

A distinctive feature of XCodec2 is its acoustic decoder architecture. Like in TS3Codec (Wu et al., 2024b), rather than using stacked convolutional upsampling blocks, it employs a Transformer backbone followed by an inverse short-time Fourier transform (iSTFT) head (Siuzdak, 2024). This design naturally suggests using the Transformer backbone outputs for computing $\mathcal{L}_{\text{guide}}$ because: (i) the Transformer contains the majority of learnable parameters in the decoder, providing sufficient capacity to benefit from self-guidance; and (ii) the subsequent iSTFT head separates the hidden features from the final waveform generation, preventing potential interference from waveform-level reconstruction losses.

The complete training objective for our enhanced codec is:

$$\mathcal{L}_{\text{total}} = \lambda_{guide}\mathcal{L}_{\text{guide}} + \mathcal{L}_{\text{semantic}} + \mathcal{L}_{\text{acoustic}} + \mathcal{L}_{\text{adv}} \quad (3)$$

which contains:

- $\mathcal{L}_{\text{guide}}$, $\lambda_{guide}$: the self-guiding feature mapping loss defined in Equation 2 (computed on the Transformer backbone outputs) and the corresponding loss weight;

- $\mathcal{L}_{\text{semantic}}$: the semantic feature MSE loss;

- $\mathcal{L}_{\text{acoustic}}$: the multi-scale Mel-spectrogram L1 loss;

- $\mathcal{L}_{\text{adv}}$: the adversarial loss from a multi-period discriminator (Kong et al., 2020) and a spectrogram discriminator (Parker et al., 2024).

## 5. Experiments and Analysis

### 5.1. Experiment Settings

**Dataset** We use the full Librispeech (Panayotov et al., 2015) training set for the training of all versions of codec

*Table 2.* Comparing reconstruction evaluation results with other existing neural codecs on the LibriSpeech test-clean dataset. (**SG** signifies the proposed self-guidance mechanism; details about each metric are included in Section A.2)

| Codecs models | Frame rate | Codebook size(s) | PESQ↑ | STOI↑ | MCD↓ | WER↓ | SIM↑ | UTMOS↑ |
|---|---|---|---|---|---|---|---|---|
| *Ground Truth* | | | 4.64 | 1.000 | 0.00 | 2.5 | 1.00 | 4.08 |
| DAC | 50Hz | 1024×8 | 2.72 | 0.940 | – | – | 0.87 | – |
| DAC | 50Hz | 1024×2 | 1.13 | 0.730 | – | – | 0.32 | – |
| WavTokenizer | 75Hz | 4096 | 2.05 | 0.886 | 4.00 | 6.8 | 0.59 | 3.89 |
| BigCodc | 80Hz | 8192 | 2.68 | 0.935 | 2.93 | 3.6 | 0.84 | 4.11 |
| WavTokenizer | 40Hz | 4096 | 1.88 | 0.868 | 4.32 | 8.0 | 0.57 | 3.77 |
| BigCodec | 40Hz | 8192 | 2.11 | 0.894 | 3.72 | 6.7 | 0.66 | 4.05 |
| XCodec2 | 50Hz | 8192 | 2.03 | 0.892 | 3.84 | 4.1 | 0.72 | **4.09** |
| XCodec2+**SG** | 50Hz | 8192 | **2.13** | **0.898** | **3.60** | **3.8** | **0.73** | 4.08 |
| TS3Codec | 40Hz | 65536 | 2.01 | 0.893 | 3.81 | 4.9 | 0.61 | 3.69 |
| TS3Codec | 40Hz | 131072 | 2.06 | 0.897 | 3.75 | 4.5 | 0.63 | 3.73 |
| TS3Codec | 50Hz | 65536 | 2.22 | 0.909 | 3.52 | 3.6 | 0.68 | 3.85 |
| TS3Codec | 50Hz | 131072 | 2.23 | 0.910 | 3.50 | 3.6 | 0.68 | 3.84 |
| XCodec2 | 50Hz | 65536 | 2.28 | 0.910 | 3.57 | 3.2 | 0.79 | 4.06 |
| XCodec2+**SG** | 50Hz | 65536 | **2.39** | **0.915** | **3.41** | **3.2** | **0.80** | **4.10** |

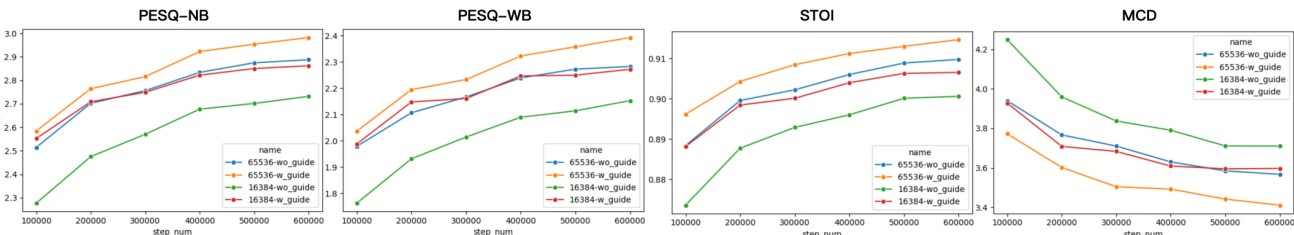

*Figure 2.* Comparison of the reconstruction performance under various settings along the training process. Horizontal axis is the training iterations. The model using a 16,384-sized codebook (red line) matches the performance of the baseline with a 4× larger codebook (65,536, blue line).

models, which comprises 960 hours of English speech audio at a sampling rate of 16kHz. For evaluation, the *test-clean* subset of LibriSpeech that contains 2620 utterances from 40 speakers is used to assess reconstruction performance.

**Implementation details** We build our neural codec model based on the offical open-source code of XCodec2 [2]. The only required modifications consist of: (i) adding an additional forward pass for the decoder during training; (ii) incorporating the proposed $\mathcal{L}_{guide}$ into the generator loss computation. Detailed configurations and sensitivity analysis on $\lambda_{guide}$ are included in Section A.1. The BigCodec model involved in the preliminary study (Section 3.2) and comparative experiments (Section 5.2) is obtained via training with the official open-source implementation [3].

**Training cost** We train all of the codec models on 8 NVIDIA GeForce RTX 4090 GPUs for 600 thousand itera-

tions. The total training time of each codec model is around 237.75 hours. Notably, the self-guidance variant incurs **negligible additional training time (<0.5%)** compared to the baseline XCodec2: 25668.0 v.s. 25783.8 (seconds per epoch). This efficiency aligns with our design in Section 4.1: the additional forward pass through the acoustic decoder requires no backward propagation, making the computational overhead minimal compared to other components (e.g., discriminators) and gradient synchronization. This demonstrates that the performance gains from self-guidance come at virtually no additional training cost.

## 5.2. Reconstruction Performance

We first evaluate the overall reconstruction performance of our proposed model against existing low-bitrate speech codecs. For DAC, WavTokenizer, and TS3Codec, we report results from papers. For BigCodec and XCodec2, we retrain models and include variations with different configurations (XCodec2 default: frame rate = 50 Hz, $|\mathcal{Q}|$ = 65,536; BigCodec default: frame rate = 80 Hz, $|\mathcal{Q}|$ = 8,192).

[2] https://github.com/zhenye234/X-Codec-2.0
[3] https://github.com/Aria-K-Alethia/BigCodec

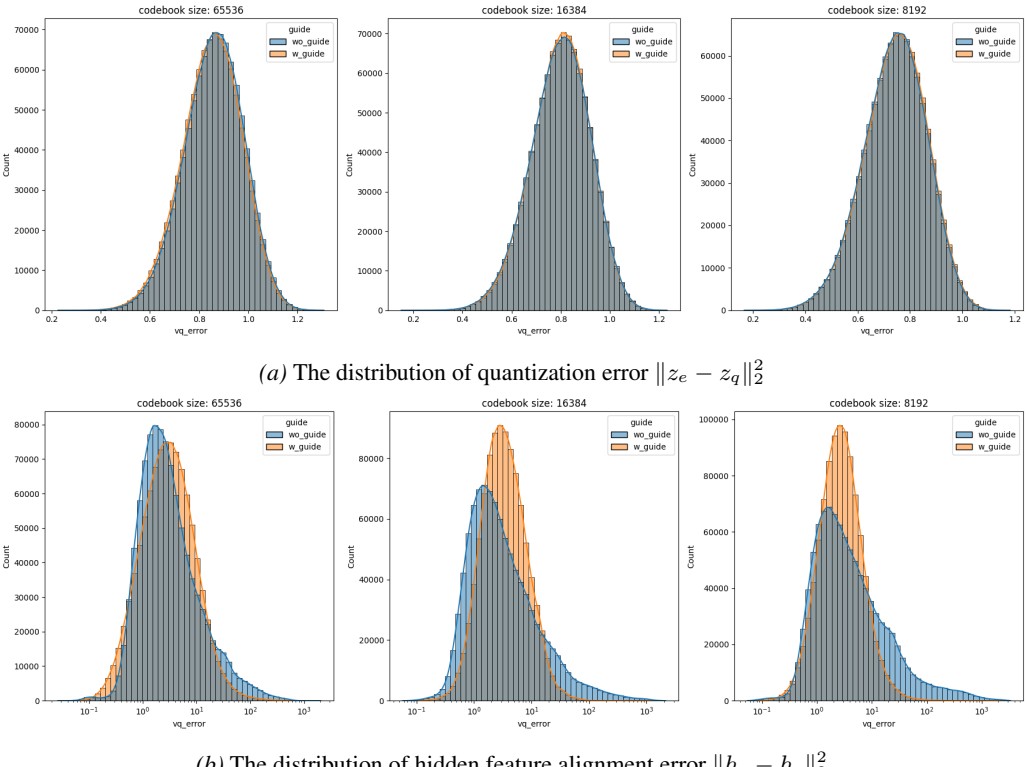

*(a)* The distribution of quantization error $\|z_e - z_q\|_2^2$

*(b)* The distribution of hidden feature alignment error $\|h_e - h_q\|_2^2$

*Figure 3.* The histogram of the quantization error $e_q$ and hidden feature alignemnt MSE on LibriSpeech test-clean dataset with the self-guidance mechanism activated (`w_guide`) or omitted (`wo_guide`) across different codebook sizes (from left to right: 65536, 16384, 8192).

As shown in Table 2, our proposed model (XCodec2 with self-guidance) achieves leading performance across most evaluation metrics. For codecs with frame rates of 40–50 Hz, our approach consistently outperforms competitors with similar codebook sizes (8,192 and below, or 65,536 and above), establishing new state-of-the-art performance for low-bitrate speech codecs. *Considering that the BigCodec (159M params) improves PESQ by* 0.15 *over TFCodec (Jiang et al., 2023) (6.37M params)with a 25x parameter increase, our method achieves a PESQ improvement of* 0.1 *over strong baselines with minimal cost (zero inference overhead, no architectural changes).*

Specifically, while the original XCodec2 with codebook size 65,536 shows competitive performance, self-guidance provides further improvements across all metrics. Reducing XCodec2's codebook size to 8,192 significantly degrades acoustic reconstruction quality (PESQ, STOI, MCD), falling behind BigCodec. However, when augmented with self-guidance, this reduced-size model surpasses BigCodec on all metrics.

In addition to the objective metrics, we conduct an AB-preference subjective evaluation to verify the perceptual quality enhancement. We randomly sampled 30 clips from

*Table 3.* Results of the AB-Preference subjective test regarding reconstruction fidelity.

| Preference | with SG | without SG | No Preference |
|---|---|---|---|
| Percentage(%) | **38.684** | 15.351 | 45.965 |

LibriSpeech test-clean. Listeners indicated their preference regarding reconstruction fidelity compared to the ground truth (GT) reference. Audios from the proposed and baseline models were anonymized and shuffled. Table 3 shows the summarized results from 38 valid evaluations, where self-guidance significantly outperforms the baseline by a $2\times$ **preference ratio**. We also include reconstruction samples in Appendix Section A.8 for improvements demonstration and failure case analysis. Corresponding audio samples are available on our demo website.[4]

### 5.3. Mechanism Analysis via Feature Alignment

To provide direct evidence that self-guidance operates by aligning the decoder's internal feature manifolds, we analyze its impact on two key error distributions: the quan-

---

[4]https://sgvqvae.github.io/sgvqvae-demo

*Table 4.* Results of manifold alignment metrics computed between teacher and student hidden features ($h_e$ and $h_q$). (**with SG** signifies whether the proposed self-guidance mechanism is applied)

| with SG | kNN Jaccard↑ | Procrustes Residuals↓ |
|---------|--------------|-----------------------|
| ✗       | 0.276        | 0.265                 |
| ✓       | **0.307**    | **0.171**             |

tization error ($e_q = \|z_e - z_q\|_2^2$) and the hidden feature alignment error ($\|h_e - h_q\|_2^2$).

**Quantization error**  Since the gradient from $\mathcal{L}_{\text{guide}}$ propagates to the encoder via straight-through estimation, we investigate whether the performance improvement originates from decoder guidance or from an implicit reduction in the quantization error itself. Figure 3a visualizes the distribution of the quantization error ($e_q$) on the test-clean set for both the baseline and self-guidance models across various codebook sizes. The nearly identical, overlapping distributions clearly indicate that *the gains from self-guidance are not attributable to a reduction in the fundamental quantization error*.

**Hidden feature alignment error**  To directly assess the mechanism of self-guidance, we analyze the hidden feature alignment error, defined as $\|h_e - h_q\|_2^2$. Its distribution, shown in Figure 3b, reveals a pronounced divergence between the baseline and our proposed method. Self-guidance significantly suppresses this error, demonstrating a successful alignment of the decoder's internal representations when processing pre-quantized versus quantized latents. This result provides direct evidence that *self-guidance enhances the decoder's robustness to quantization error by explicitly aligning its internal feature manifolds*, thereby improving reconstruction fidelity. The effect becomes increasingly pronounced as the inherent quantization error grows with smaller codebook sizes, further underscoring the utility of our method under **more constrained quantization bottlenecks**. Detailed statistical summaries for both error distributions are provided in Section A.7.

### 5.4. Validation on Decoder Manifold Alignment

In addition to the MSE metrics, we include further statistical analyses and visualization to provide stronger evidence for manifold alignment. Two complementary manifold alignment metrics are incorporated to the hidden features.

1. **kNN Jaccard similarity (higher is better)**: Measures if the same points are neighbors across the two manifolds (teacher vs. student). (Heinrichs et al., 2023)

2. **Procrustes residuals (lower is better)**: Measures overall geometric similarity after optimal rotation. (Williams et al., 2021)

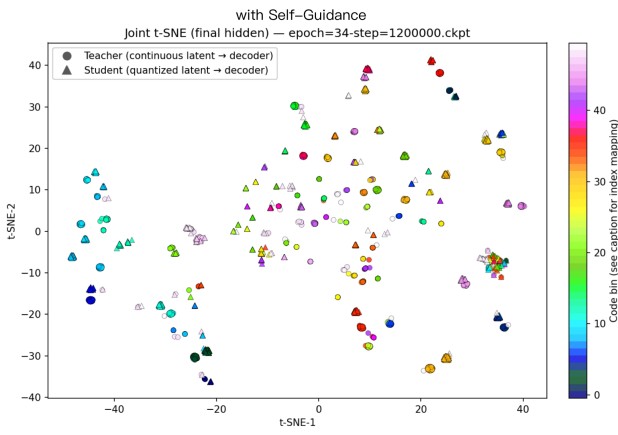

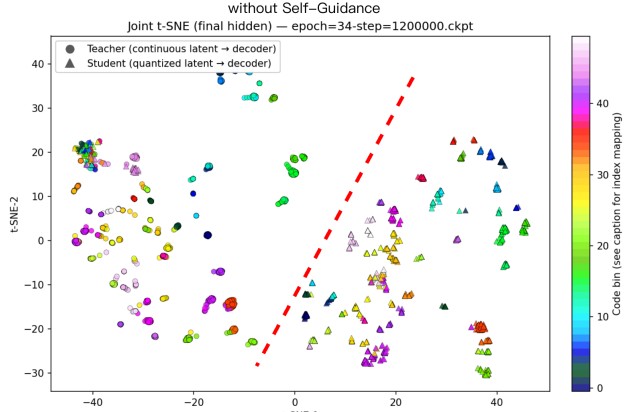

*Figure 4.* The t-SNE visualization result of the decoder hidden features from the top-50 most frequent quantized tokens (each token ID signified in a unique color). Round markers stand for teacher feature $h_e$, while triangle markers stand for student feature $h_q$. In the baseline approach (lower), features from the two branches separate into distinct halves of the latent space (red dashed line).

As shown in Table 4, self-guidance significantly improves both local neighborhood structure (kNN Jaccard) and global geometric alignment (Procrustes), supporting the manifold alignment claim beyond simple MSE reduction.

**t-SNE Visualiation**  We also compare the t-SNE visualization (Van der Maaten & Hinton, 2008) results between self-guidance and the baseline apporach in Figure 4. With self-guidance activated, the hidden features from both the continuous teacher (circles) and quantized student (triangles) cluster by token ID (color, 50 most common tokens), indicating that SG preserves discriminative, token-specific information and does not induce feature collapse. While in the baseline approach, features from the two branches separate into distinct halves of the latent space (red dashed line), showing baseline manifold misalignment. This is not a collapse, but it visualizes the dual-path inconsistency that self-guidance is designed to mitigate.

*Table 5.* Reconstruction evaluation results of the proposed neural speech codec across different codebook sizes. (**with SG** signifies whether the proposed self-guidance mechanism is applied)

| Codebook size | with SG | PESQ-WB↑ | PESQ-NB↑ | STOI↑ | MCD↓ | WER↓ | SIM↑ | UTMOS↑ |
|---|---|---|---|---|---|---|---|---|
| *Ground Truth* | | 4.64 | 4.54 | 1.000 | 0.00 | 2.49 | 1.00 | 4.08 |
| 8192 | ✗ | 2.03 | 2.59 | 0.892 | 3.84 | 4.08 | 0.72 | **4.09** |
| 8192 | ✓ | **2.13** | **2.69** | **0.898** | **3.79** | **3.77** | **0.73** | 4.08 |
| 16384 | ✗ | 2.15 | 2.73 | 0.901 | 3.71 | **3.47** | 0.76 | 3.98 |
| 16384 | ✓ | **2.27** | **2.86** | **0.907** | **3.70** | 3.53 | **0.77** | **4.08** |
| 65536 | ✗ | 2.28 | 2.89 | 0.910 | 3.57 | 3.23 | 0.79 | 4.06 |
| 65536 | ✓ | **2.39** | **2.98** | **0.915** | **3.41** | **3.15** | **0.80** | **4.10** |

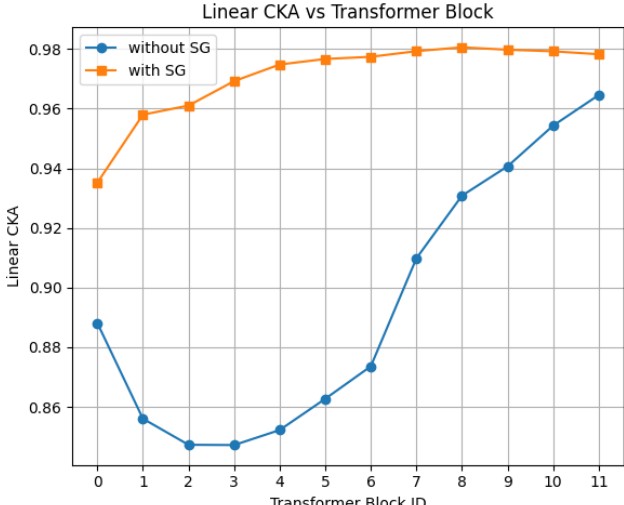

*Figure 5.* Blockwise linear CKA results between the teacher and student branch. Higher values reveal better alignment. The proposed SG (orange) incurs substantial alignment improvements throughout the decoder.

*Table 6.* Reconstruction evaluation results when applying SG under various codec settings. The more comprehensive results are included in the Appendix Section A.3, A.4 and A.5. (**with SG** signifies whether the proposed self-guidance is applied)

| with SG | PESQ↑ | STOI↑ | MCD↓ | SIM↑ | UTMOS↑ |
|---|---|---|---|---|---|
| *SimVQ: Projector-based VQ ($|\mathcal{Q}| = 16384$)* | | | | | |
| ✗ | 2.10 | 0.900 | 3.63 | 0.75 | 3.85 |
| ✓ | **2.17** | **0.904** | **3.56** | **0.76** | **3.93** |
| *ResidualFSQ: Multi-codebook VQ ($N_{VQ} = 2, |\mathcal{Q}_i| = 1024$)* | | | | | |
| ✗ | 1.75 | 0.878 | 4.21 | 0.65 | **3.39** |
| ✓ | **1.86** | **0.880** | **4.08** | **0.67** | **3.41** |
| *BigCodec: CNN-based codec ($|\mathcal{Q}| = 8192$)* | | | | | |
| ✗ | 1.67 | 0.860 | 4.32 | 0.46 | 3.57 |
| ✓ | **1.77** | **0.866** | **4.22** | **0.51** | **3.80** |

**Blockwise Alignment Analysis** Beyond the final hidden states of the decoder Transformer backbone, another blockwise manifold alignment analysis is performed across all 12 transformer blocks in the decoder, where the **linear CKA (Centered Kernel Alignment, higher is better)** (Kornblith et al., 2019) is computed between the outputs from the teacher and student branches. As shown in Figure 13, despite that self-guidance only applies the feature mapping loss on the final hidden state of the Transformer backbone, it substantially improves alignment throughout the decoder. Detailed values on each block are included in Section A.6

### 5.5. Generalization Across Diverse VQ-VAE Configs

This section evaluates the general applicability of self-guidance by applying it to VQ-VAE models with varied architectural components. The consistent performance im-

provements observed across these modifications underscore its potential as *a general-purpose enhancement for high-fidelity neural speech codecs*.

**Varying Codebook Size** We evaluate self-guidance with codebook sizes of 8,192, 16,384, and 65,536. As shown in Table 5, it improves performance across nearly all metrics in each setting. A key finding, illustrated in Figure 2, is that with self-guidance, a model using a 16,384-sized codebook matches or exceeds the performance of the baseline XCodec2 with a 4× larger codebook (65,536) on several key metrics, highlighting its efficiency.

**Varying Vector Quantizer Type** To demonstrate independence from a specific quantization algorithm, we apply self-guidance to XCodec2 equipped with two alternative quantizers: SimVQ (a projected codebook method) and Residual FSQ (a multi-codebook quantizer). As shown in Table 6, consistent gains are observed in both cases, confirming the method's adaptability. Detailed settings results are provided in Appendix Sections A.3 and A.4.

**Varying Decoder Architecture** Finally, we validate self-guidance on BigCodec, a state-of-the-art codec with a decoder based on stacked transposed convolutional layers for progressive upsampling—a stark contrast to XCodec2's transformer-based, fixed-resolution decoder. As shown in Table 6, self-guidance yields noticeable improvements across all metrics on this architecture, demonstrating its effectiveness beyond a specific network backbone. Detailed settings and results are provided in Appendix Section A.5.

### 5.6. Downstream Auto-Regressive TTS

Building on our finding that self-guidance enables smaller codebooks to achieve performance comparable to larger ones (Figure 2), we evaluate its impact on downstream autoregressive text-to-speech (TTS) synthesis. We hypothesize that reduced codebook size simplifies the language modeling task, potentially improving final TTS quality.

We conduct rapid TTS experiments using a Qwen2.5-0.5B causal LLM backbone (Qwen et al., 2025) trained on LibriTTS-R (Koizumi et al., 2023). Input text is phonemized before training. Models are supervised fine-tuned for phoneme-to-audio-token generation for 85 epochs. For inference, we use the continual synthesis approach from VALL-E (Wang et al., 2023), providing phoneme sequences and first 3-second audio tokens as prompts for continuation generation.

*Table 7.* Downstream text-to-speech continuation performance on the LibriTTS test-clean dataset.

| Codec model | Codebook size | UTMOS↑ | WER↓ | SIM↑ |
|---|---|---|---|---|
| XCodec2 | 65536 | 3.33 | 33.03 | 0.58 |
| XCodec2+**SG** | 65536 | 3.39 | 35.07 | 0.58 |
| XCodec2 | 16384 | 3.51 | 28.78 | 0.56 |
| XCodec2+**SG** | 16384 | **3.58** | **28.02** | 0.58 |

As shown in Table 7, the TTS performance strongly correlates with codebook size, where models using a 16,384 codebook significantly outperform those with a 65,536 codebook. This **aligns with our hypothesis in Section 1 that a large flat audio token vocabulary is harmful to the downstream LLMs**. Within this context, self-guidance yields the best performance at the 16,384 size. At the 65,536 size, the results of self-guidance are mixed, as the TTS model is primarily hampered by the fundamental language modeling difficulty of the large codebook, which overshadows the fidelity improvement from self-guidance.

### 6. Conclusion

We propose self-guidance, a novel training mechanism for VQ-VAE-based neural speech codecs that enhances decoder robustness to quantization artifacts. By aligning the decoder's outputs for quantized and continuous latent representations through an additional feature-mapping loss, our method improves reconstruction fidelity without modifying the inference process. Experiments demonstrate that self-guidance consistently enhances performance across various VQ-VAE configurations, enabling comparable quality with 4× smaller codebooks. Downstream TTS results confirm that this reduction simplifies language modeling for LLMs, improving synthesis quality. Our approach provides a general, effective, and efficient solution to mitigate quantization errors, advancing high-fidelity neural speech compression.

**Limitations and Future Work** While self-guidance significantly mitigates the reconstruction degradation caused by quantization error, it does not eliminate such artifacts entirely; residual distortion may persist in certain cases due to training dynamics. The current validation is limited to neural speech codecs, and future work could extend self-guidance to general audio codecs (e.g., sound effects, music) and non-audio VQ-VAEs (e.g., the image domain). Additionally, the downstream LLM/TTS results, though directionally positive, remain preliminary. Scaling these experiments would yield more comprehensive evidence of real-world benefit.

### Acknowledgements

This work is supported by the National Natural Science Foundation of China (62076144) and the Major Key Project of Peng Cheng Laboratory (PCL2024A08).

### Impact Statement

This work presents research on neural audio codecs, which have significant potential for positive applications in speech compression, communication, and generative modeling. However, we acknowledge several ethical considerations:

1. **Positive Impacts:** Our method enables higher-quality audio compression at lower bitrates, which could improve accessibility and efficiency in telecommunication, hearing assistance devices, and low-bandwidth applications. The reduction in codebook complexity may also decrease computational requirements for downstream applications.

2. **Potential Misuse:** Like other audio generation technologies, neural codecs could potentially be misused for creating deepfake audio or other deceptive content. However, our work focuses specifically on reconstruction quality rather than generative capabilities. The codec itself does not generate novel content without being integrated into a full generative system.

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

# A. Appendix

## A.1. Model Configuration

The detailed model configuration and loss weights are listed in Table 8. Most of the configurations follows the default configuration of XCodec2. Specifically, the weight of the proposed self-guidance feature mapping loss weight $\lambda_{guide}$ is selected from the best of [1, 5, 10, 15], according to the overall reconstruction performance in test trials.

*Table 8.* Model configurations

| Configuration entry | Value |
|---|---|
| Acoustic encoder hidden dim | 1024 |
| Acoustic encoder convoution blocks | 5 |
| Acoustic encoder up ratio | [2, 2, 4, 4, 5] |
| Acoustic decoder hidden dim | 1024 |
| Acoustic decoder Transformer layers | 12 |
| Semantic encoder hidden dim | 1024 |
| Semantic decoder hidden dim | 1024 |
| FSQ scales (codebook size = 65536) | [4, 4, 4, 4, 4, 4, 4, 4] |
| FSQ scales (codebook size = 16384) | [4, 4, 4, 4, 4, 4, 4] |
| FSQ scales (codebook size = 8192) | [4, 4, 4, 4, 4, 4, 2] |
| loss weight $\lambda_{semantic}$ | 5.0 |
| loss weight $\lambda_{acoustic}$ | 15.0 |
| loss weight $\lambda_{adv}$ | 1.0 |
| loss weight $\lambda_{guide}$ (codebook size = 65536) | 5.0 |
| loss weight $\lambda_{guide}$ (codebook size = 16384) | 10.0 |
| loss weight $\lambda_{guide}$ (codebook size = 8192) | 10.0 |
| batch size | 16 |
| optimizer | AdamW |
| optimizer betas | [0.8, 0.9] |
| learning rate warmup steps | 1000 |
| learning rate decay steps | 500000 |
| learning rate min value | 2e-5 |
| learning rate max value | 1e-4 |

**Sensitivity analysis on $\lambda_{guide}$**   A sensitivity analysis was conducted on the guidance loss weight ($\lambda_{guide}$) to assess its impact on model performance. The results that we draw from XCodec2 with codebook size 16384 at the 200k training step, as detailed in Table 9, indicate a clear trend:

1. When the weight is too small ($\lambda_{guide} = 1$), the guidance effect is negligible, yielding results similar to the baseline.

2. An optimal range is observed between $\lambda_{guide} = 5$ and 10, where the method achieves significant and robust improvements across nearly all metrics, indicating relative insensitivity to small changes within this window.

3. Conversely, values of $\lambda_{guide} \geq 15$ cause the auxiliary loss to dominate the training objective, leading to a performance drop.

This analysis confirms a stable optimal range and validates our selection of $\lambda_{guide} = 10$, providing clear practical guidance for future implementations.

## A.2. Evaluation Metrics of Reconstruction

We evaluate acoustic fidelity, intelligibility, and naturalness of the speech audio reconstructed by neural codecs using the following metrics:

*Table 9.* Reconstruction evaluation results of different $\lambda_{guide}$ values at 200k training steps, with XCodec2 codebook size fixed at 16384.

| $\lambda_{guide}$ | PESQ-WB↑ | PESQ-NB↑ | STOI↑ | MCD↓ | WER↓ | SIM↑ | UTMOS↑ |
|---|---|---|---|---|---|---|---|
| 0. (baseline) | 1.9309 | 2.4747 | 0.8877 | 3.9591 | 4.08 | 0.7088 | 3.6752 |
| 1 | 1.9219 | 2.4533 | 0.8881 | 3.9527 | 3.87 | 0.7148 | 3.7266 |
| 5 | 2.1166 | 2.6705 | 0.8977 | **3.6796** | **3.56** | **0.7488** | 3.8352 |
| 10 | **2.1474** | **2.7082** | **0.8984** | 3.7086 | 3.87 | 0.7428 | **3.8395** |
| 15 | 2.0409 | 2.6074 | 0.8936 | 3.7796 | 3.95 | 0.7374 | 3.7627 |
| 50 | 1.9462 | 2.4904 | 0.8883 | 3.9035 | 4.18 | 0.7073 | 3.7878 |
| 100 | 1.8779 | 2.4312 | 0.8822 | 3.9648 | 4.40 | 0.6845 | 3.7039 |

**Perceptual Evaluation of Speech Quality (PESQ)**   PESQ (Rix et al., 2001) compares degraded and reference speech to predict human-perceived quality. We use a Python implementation [5] to compute wide-band (PESQ-WB) and narrow-band (PESQ-NB) scores, where higher values indicate better quality.

**Mel Cepstral Distortion (MCD)**   MCD measures the difference between mel-frequency cepstral coefficients (MFCCs), a standard metric for speech synthesis quality.

**Short-Time Objective Intelligibility (STOI)**   STOI (Taal et al., 2011) evaluates speech intelligibility by comparing temporal envelopes of clean and degraded signals, with scores ranging from 0 (unintelligible) to 1 (perfect intelligibility).

**Word Error Rate (WER)**   WER is calculated using a HuBERT (Hsu et al., 2021) speech recognition model finetuned on Librispeech [6], reporting percentage errors in transcribed words.

**Speaker Similarity (SIM)**   Speaker characteristics are evaluated via cosine similarity between original and reconstructed utterances, using a WavLM-large (Chen et al., 2022)-based speaker verification model [7].

**UTMOS**   UTMOS (Saeki et al., 2022) predicts Mean Opinion Score (MOS) for speech naturalness, with scores from 1 (poor) to 5 (excellent). We use a pretrained UTMOS strong model [8].

### A.3. Generalization Experiment on SimVQ

To assess generalization across quantizer types, we replace the default FSQ quantizer in XCodec2 with SimVQ (VQGAN-FC suffered from codebook collapse and produced unintelligible results). Table 10 shows that self-guidance consistently improves performance with SimVQ, reproducing the minor WER degradation observed with FSQ. Since we only observe slight WER rise of 0.06% at codebook size 16384, there appears to be no systematic trend for sacrificing intelligibility, especially given the consistent gains in the spectral intelligibility metric STOI.

*Table 10.* Reconstruction evaluation results of the proposed neural speech codec across different types of vector quantizers (XCodec2 adopts FSQ by default), with codebook size fixed at 16384. (**with SG** signifies whether proposed self-guidance mechanism is applied).

| Quantizer | with SG | PESQ-WB↑ | PESQ-NB↑ | STOI↑ | MCD↓ | WER↓ | SIM↑ | UTMOS↑ |
|---|---|---|---|---|---|---|---|---|
| *Ground Truth* | | 4.64 | 4.54 | 1.000 | 0.00 | 2.49 | 1.00 | 4.08 |
| FSQ | ✗ | 2.15 | 2.73 | 0.901 | 3.71 | **3.47** | 0.76 | 3.98 |
| FSQ | ✓ | **2.27** | **2.86** | **0.907** | **3.60** | 3.53 | **0.77** | **4.08** |
| SimVQ | ✗ | 2.10 | 2.67 | 0.900 | 3.63 | **3.59** | 0.75 | 3.85 |
| SimVQ | ✓ | **2.17** | **2.74** | **0.904** | **3.56** | 3.63 | **0.76** | **3.93** |

---

[5] https://github.com/ludlows/PESQ
[6] https://huggingface.co/facebook/hubert-large-ls960-ft
[7] https://github.com/microsoft/UniSpeech/tree/main/downstreams/speaker_verification
[8] https://github.com/tarepan/SpeechMOS

## A.4. Generalization Experiment on Residual FSQ

To evaluate the general applicability of the proposed self-guidance (SG) loss beyond single-codebook models, we integrated it into a Residual FSQ architecture. The model was configured with two FSQ layers, each employing a scale of [4,4,4,4,4] (resulting in a codebook size of 1024 per layer). When integrated into the XCodec2 framework at a 50 Hz frame rate, this configuration yields a total bitrate of 1000 bps. The model was trained for 100k steps. As shown in Table 11, applying the SG loss led to consistent improvements across all objective metrics. This demonstrates that the self-guidance principle is effective not only for standard single-codebook VQ but also for multi-stage residual quantization paradigms.

*Table 11.* Reconstruction evaluation results on XCodec2 with Residual FSQ at 100k step, with codebook size fixed at 1024x2. (**with SG** signifies whether the proposed self-guidance mechanism is applied).

| with SG | PESQ-WB↑ | PESQ-NB↑ | STOI↑ | MCD↓ | WER↓ | SIM↑ | UTMOS↑ |
|---|---|---|---|---|---|---|---|
| ✗ | 1.7539 | 2.2503 | 0.8768 | 4.2158 | 4.30 | 0.6466 | 3.3923 |
| ✓ | **1.8594** | **2.4154** | **0.8802** | **4.0819** | **4.18** | **0.6747** | **3.4105** |

## A.5. Generalization Experiment on BigCodec

To further validate the generality of our method across different model architectures, we applied the self-guidance loss to the BigCodec framework. We adapted the open-source BigCodec model to operate at a 40 Hz frame rate—comparable to our other experiments—by introducing additional down- and up-sampling blocks in the encoder and decoder. This follows the setting of 40Hz BigCodec presented in Table 2. The model used the default codebook size of 8192, with the VQ mechanism upgraded from vanilla VQ-FC to FSQ for better training stability. Specifically, unlike the Transformer blocks in the XCodec2 decoder, which consistently operate on the same feature frame rate (50Hz), each of the convolutional upsampling blocks in the BigCodec decoder operates on a different feature frame rate (gradually upsampled from 40Hz to 16kHz). Thus, in addition to the self-guidance feature mapping loss on the final block, we also insert self-guidance loss at the end of each previous upsampling block in the BigCodec decoder. For the rest of the training configuration, we follow the official implementation and conduct the evaluation when the training process reaches 100k iterations.

*Table 12.* Reconstruction evaluation results BigCodec (framerate 40Hz) at 100k step, with codebook size fixed at 8192. (**with SG** signifies whether proposed self-guidance mechanism is applied).

| with SG | PESQ-WB↑ | PESQ-NB↑ | STOI↑ | MCD↓ | WER↓ | SIM↑ | UTMOS↑ |
|---|---|---|---|---|---|---|---|
| ✗ | 1.6740 | 2.1795 | 0.8601 | 4.3179 | 11.86 | 0.4634 | 3.5694 |
| ✓ | **1.7650** | **2.3037** | **0.8655** | **4.2161** | **10.98** | **0.5072** | **3.8040** |

The evaluation results confirm that self-guidance provides a clear and consistent performance gain on this architecture, which features an RNN/CNN-based decoder fundamentally different from the Transformer-based decoder of XCodec2. This result robustly confirms that our method is a general-purpose technique for enhancing decoder robustness, independent of the specific backbone architecture.

## A.6. Blockwise Alignment Measurement

In addition to the linear CKA values plotted in Figure 5, we also compute the Spearman correlation of pairwise distances in hidden space (Kriegeskorte et al., 2008) between the teacher and student. Both values are presented in Table 13, where the proposed self-guidance consistently promotes decoder intermediate latent manifold alignment.

## A.7. Detailed Statistics of the Error Distributions in Figure 3

Here we provide the detailed statistics of the distributions presented in Figure 3. Table 14 contains statistics of quantization error distribution across different codebook sizes (Figure 3a), while Table 15 contains statistics of the hidden feature alignment MSEs (Figure 3b), respectively. These results align with the illustrative demonstrative, where the proposed self-guidance makes little shift to the quantization error distribution, but significantly reduces both the error level and dispersion in hidden feature alignment MSE. Specifically, we also observe that the latter effect becomes more obvious as the codebook size further shrinks.

*Table 13.* Blockwise linear CKA and Spearman correlation values between the teacher and student branch.

| Block | with SG | Spearman↑ | linear CKA↑ |
|---|---|---|---|
| 0 | ✗ | 0.747 | 0.888 |
| 0 | ✓ | **0.920** | **0.935** |
| 1 | ✗ | 0.690 | 0.856 |
| 1 | ✓ | **0.931** | **0.958** |
| 2 | ✗ | 0.706 | 0.847 |
| 2 | ✓ | **0.927** | **0.961** |
| 3 | ✗ | 0.730 | 0.847 |
| 3 | ✓ | **0.925** | **0.969** |
| 4 | ✗ | 0.741 | 0.852 |
| 4 | ✓ | **0.927** | **0.975** |
| 5 | ✗ | 0.737 | 0.863 |
| 5 | ✓ | **0.929** | **0.977** |
| 6 | ✗ | 0.735 | 0.874 |
| 6 | ✓ | **0.927** | **0.977** |
| 7 | ✗ | 0.807 | 0.910 |
| 7 | ✓ | **0.923** | **0.979** |
| 8 | ✗ | 0.828 | 0.931 |
| 8 | ✓ | **0.925** | **0.981** |
| 9 | ✗ | 0.830 | 0.941 |
| 9 | ✓ | **0.927** | **0.980** |
| 10 | ✗ | 0.864 | 0.954 |
| 10 | ✓ | **0.934** | **0.979** |
| 11 | ✗ | 0.933 | 0.965 |
| 11 | ✓ | **0.946** | **0.978** |

*Table 14.* Quantization Error distributions across different codebook sizes.

| Codebook size | With self-guidance | Error mean | Error std |
|---|---|---|---|
| 65536 | No | 0.858 | 0.120 |
| | **Yes** | 0.851 | 0.121 |
| 16384 | No | 0.798 | 0.121 |
| | **Yes** | 0.799 | 0.120 |
| 8192 | No | 0.741 | 0.120 |
| | **Yes** | 0.744 | 0.121 |

*Table 15.* Hidden MSE distributions across different codebook sizes.

| Codebook size | With self-guidance | Error mean | Error std |
|---|---|---|---|
| 65536 | No | 9.439 | 29.203 |
| | **Yes** | **5.854** | **13.712** |
| 16384 | No | 13.551 | 59.137 |
| | **Yes** | **4.958** | **5.863** |
| 8192 | No | 23.605 | 109.458 |
| | **Yes** | **4.197** | **6.865** |

## A.8. Reconstruction Samples

Here we provide the reconstruction samples on the LibriSpeech *test-clean* dataset to reveal the typical quantization artifacts in the generated audio caused by quantization error, including smeared harmonics (Figure 6), pitch spikes (Figure 7), and oversmoothed harmonics (Figure 8).

While self-guidance significantly lowers the overall frequency of such artifacts, it does not eliminate them entirely. Due to training dynamics, the proposed approach may still present some artifacts in certain cases. Figure 9 presents a case which contains a depressed pitch in the starting word of the reconstructed utterance.

All of the corresponding audio for these samples could be listened to on our demo website.[9]

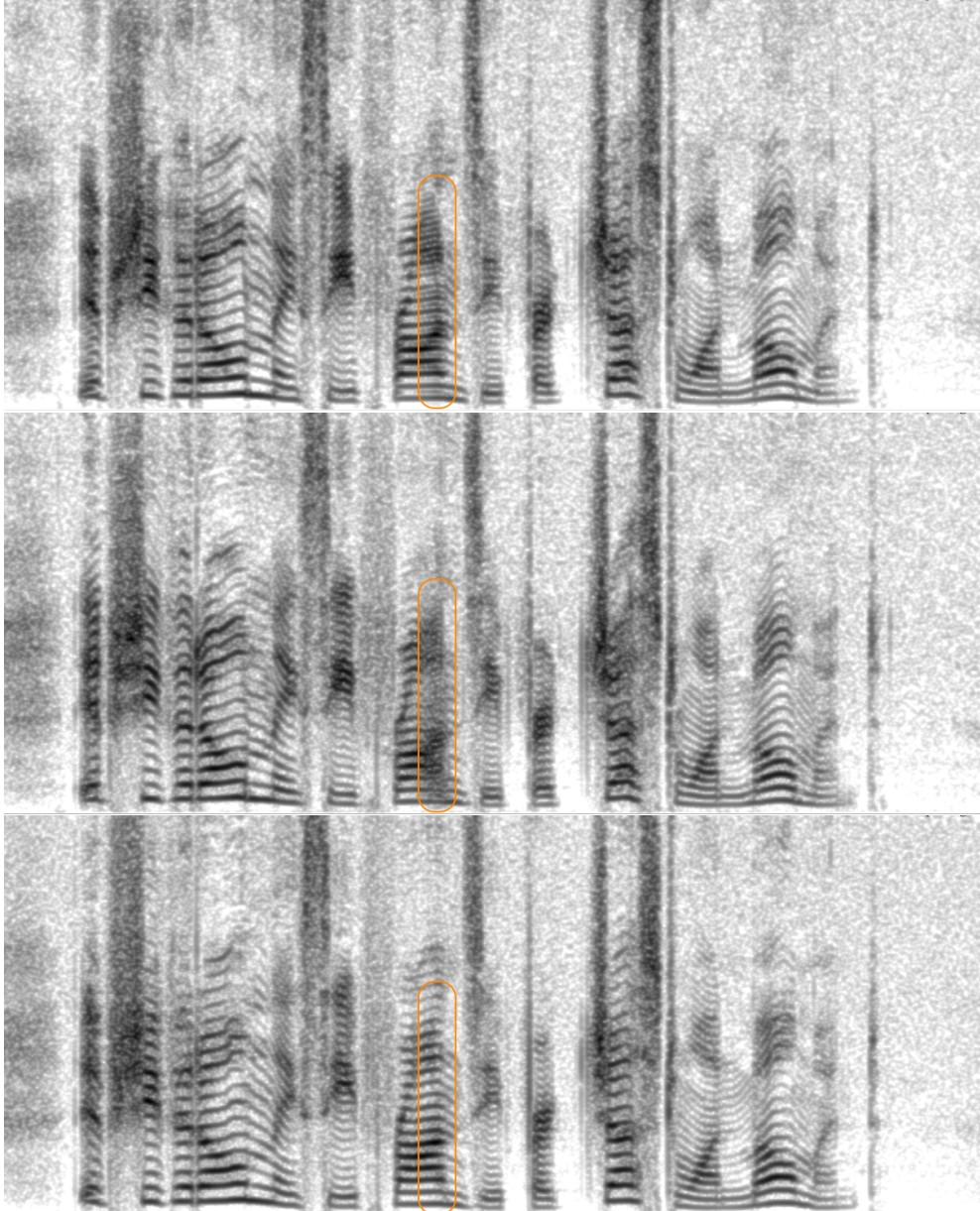

*Figure 6.* Audio spectrograms of LibriSpeech *test-clean* set sample 237-126133-0004. From top to bottom: ground truth, reconstructed result from vanilla XCodec (baseline), and reconstructed result from XCodec2 with self-guidance, respectively. The baseline system generates **smeared harmonics** in the segment signified by the orange rectangle.

---

[9]https://sgvqvae.github.io/sgvqvae-demo

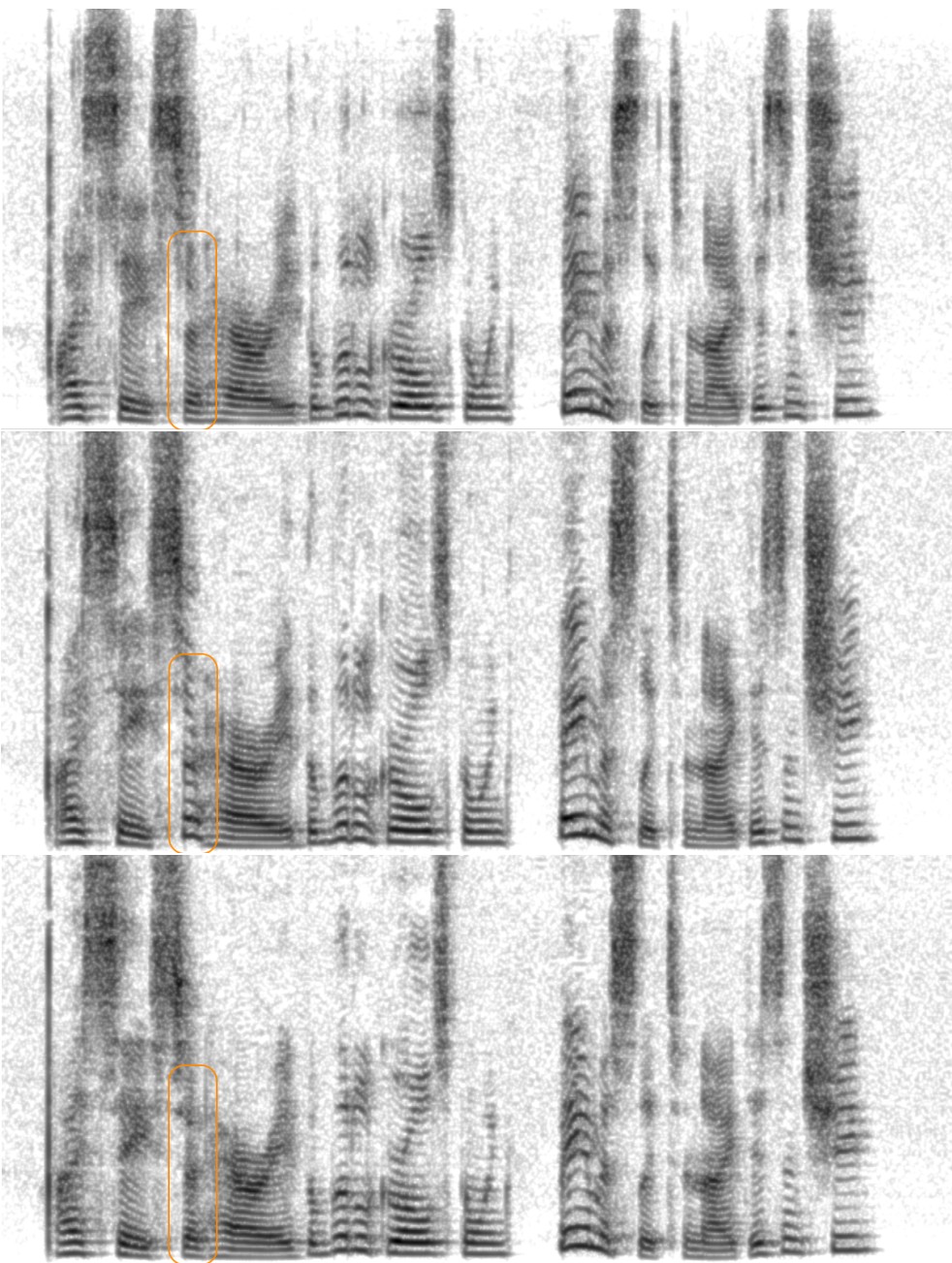

*Figure 7.* Audio spectrograms of LibriSpeech *test-clean* set sample `4446-2271-0012`. From top to bottom: ground truth, reconstructed result from vanilla XCodec (baseline), and reconstructed result from XCodec2 with self-guidance, respectively. The baseline system generates **pitch spike** in the segment signified by the orange rectangle.

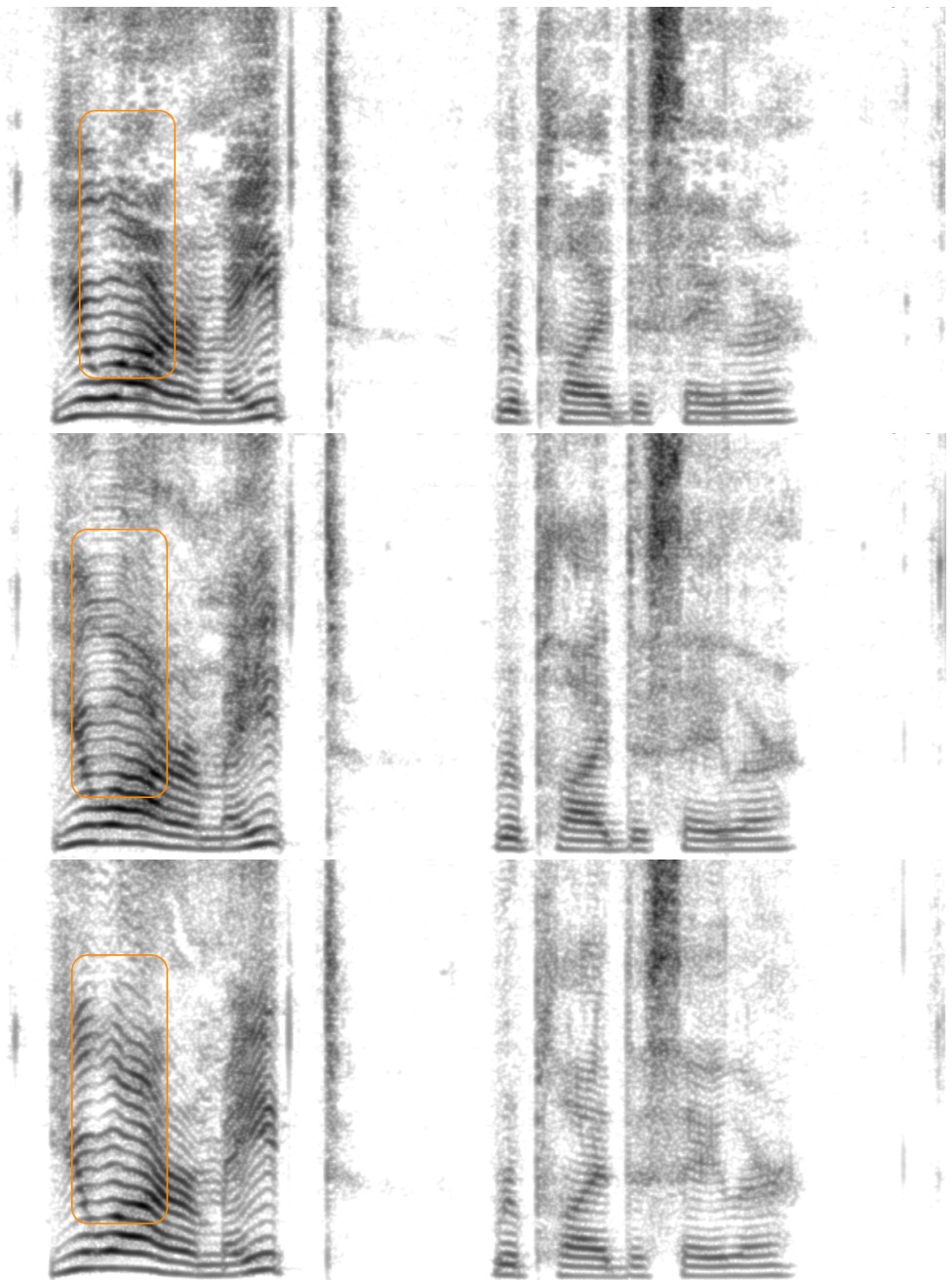

*Figure 8.* Audio spectrograms of LibriSpeech *test-clean* set sample `8555-284449-0009`. From top to bottom: ground truth, reconstructed result from vanilla XCodec (baseline), and reconstructed result from XCodec2 with self-guidance, respectively. The baseline system generates **oversmoothed harmonics** in the segment signified by the orange rectangle.

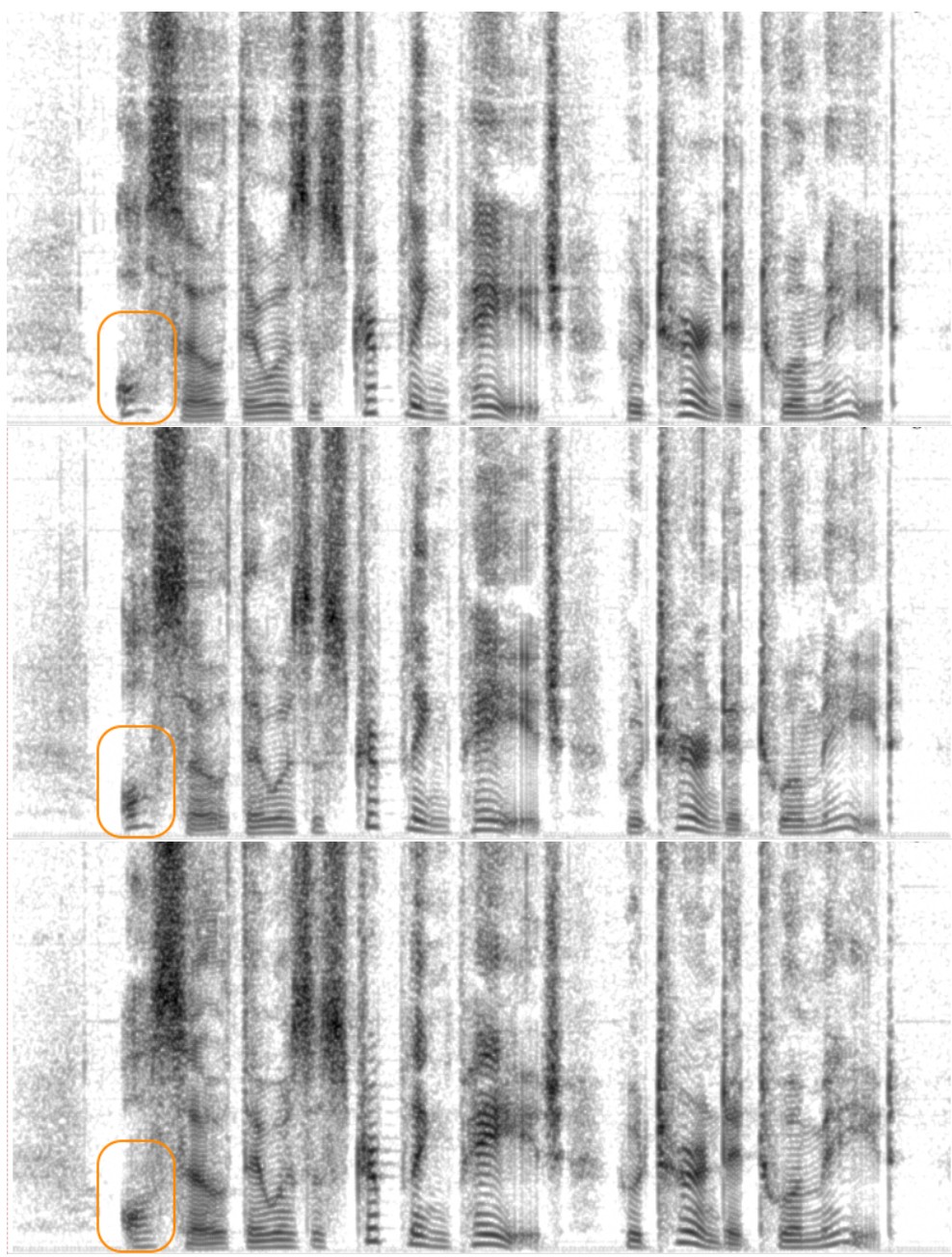

*Figure 9.* Audio spectrograms of LibriSpeech *test-clean* set sample `61-70968-0004`. From top to bottom: ground truth, reconstructed result from vanilla XCodec (baseline), and reconstructed result from XCodec2 with self-guidance, respectively. Due to training dynamics, the proposed approach may still present some artifacts in certain cases. The reconstructed audio from the proposed approach here presents a **depressed pitch** in the starting word "also" of the utterance.

