# OpenReview forum: "Self-Guidance: Enhancing Neural Codecs via Decoder Manifold Alignment"
_ICML.cc/2026/Conference — ICML 2026 regular_

### Official Review · Reviewer_SQup · 2026-03-03

**Soundness:** 2
**Presentation:** 2
**Significance:** 2
**Originality:** 2
**Overall Recommendation:** 5
**Confidence:** 5

**Summary:**

The paper proposes to apply self-distillation/guidance on the decoder of a VQ-VAE to make it robust to quantization errors, rather than reducing quantization error itself. The proposed method is experimentally validated in the context of neural audio codecs. Ablation studies across various codec architectures and quantization methods show decent performance gains.

**Compliance With Llm Reviewing Policy:**

Affirmed.

**Final Justification:**

The inclusion of the subjective evaluatio, highlighting the failure case along with the evaluation of the quantized and pre-quantized latents for decoding will improve the strength of the paper. I will raise my score to Accept.

**Key Questions For Authors:**

See the weaknesses above.

**Limitations:**

Limitations have not been explicitly included. It would be good to include the limitations of this work.

**Strengths And Weaknesses:**

**Strengths:**
- The idea is simple and intuitive and is shown to work well through experimental validation in neural audio codecs.

**Weaknesses:**
- The title of the paper seems to be claiming a lot.
  - Self-guidance/distillation as a concept is not new and has been extensively explored in previous works.
  - Decoder Manifold alignment is a very strong wording and not substantially validated, apart from showing that the error distribution of the decoders outputs are different when guidance is used or not.
  - The title reads like a audio codec paper, but the core methodology and method illustration is written in context of a generic VQ-VAE. And the experiments are only conducted on audio codecs. A small experiment on images would also provide completeness.
- The proposed methods shows consistent gains in the metrics, but it is questionable whether that reflects in actual perceptual quality improvements. No subjective evaluation is performed. Even for the downstream TTS task, improvements are modest, especially the WER seems to be very high.
- The statement in Sec. 5.5 regarding the downsides of large flat vocabulary for LLMs is validated, but it does not necessarily address or motivate the proposed method directly.  If large flat vocabulary is bad, then won't using RVQ or product quantization like strategies lead to better performance than simply making the decoder robust?

---

> ### Author Rebuttal · Authors · 2026-03-31
>
> Thank you for your candid and detailed review. We appreciate your recognition that our idea is "simple and intuitive" and shows "decent performance gains." We take your concerns seriously and address each below.
>
> > **1. Regarding the originality of self-guidance**
>
> As discussed in the related works, we acknowledge that self-distillation is a well-established concept. However, our contribution is a **novel mechanism** specifically designed to solve the **discretization bottleneck** in VQ-VAEs, rather than a generic distillation application.
>
> Key distinctions:
> 1. **Internal, single-pass guidance**: Traditional distillation requires a pre-trained teacher or EMA. Our method uses the continuous pre-quantized latent from the *exact same forward pass*, requiring no pre-training and virtually no extra compute.
> 2. **Novel targeted problem**: We use the continuous manifold to force the decoder to become robust to the spatial precision lost during quantization.
> 3. **Application domain**: To our knowledge, we are the first to demonstrate that treating quantization error as a decoder-side robustness problem—rather than trying to reduce the error at the bottleneck itself—is a highly effective strategy for neural audio codecs.
>
> > **2. Regarding justification for manifold alignment**
>
> In addition to MSE metrics, we conducted two new analyses providing stronger evidence for manifold alignment. (We will include these in the camera-ready version.)
>
> 1. **Two complementary manifold alignment metrics** on hidden features. Self-guidance significantly improves both local neighborhood structure (kNN Jaccard) and global geometric alignment (Procrustes).
>
> 2. **A block-wise analysis of manifold alignment** across all 12 transformer blocks in the decoder. Using Spearman correlation and linear CKA, we show that self-guidance substantially improves alignment at every single layer.
>
> Due to the content length limit, we kindly refer you to our response to _Reviewer xJ8v (point 1)_ for full quantitative tables, where a similar question is addressed.
>
> > **3. Regarding the presentation context**
>
> We thank the reviewer for this observation. The core motivation of this paper is to address quantization artifacts specifically in neural audio codecs, as reflected in our title, abstract, methodology, and experimental design.
>
> Section 2 and Figure 1 were written in a generic VQ-VAE context to provide a high-level mathematical illustration before diving into audio architectures. To align this framing with the rest of the paper, we will revise Section 2 and Figure 1 in the camera-ready version to explicitly use concrete audio-specific terminology.
>
> Since our claims and evaluations are focused on audio codecs, image-domain experiments fall outside the scope of this work, though we will note cross-domain applications as an exciting future direction in the revised conclusion.
>
> > **4. Regarding perceptual quality improvements**
>
> We acknowledge the importance of perceptual quality, which is addressed via:
> 1. **Objective Proxy:** We reported UTMOS, a neural MOS predictor trained to correlate with human ratings. It is a standard, reliable proxy for perceptual quality in recent speech codec literature (e.g., StableCodec/TAAE, TS3Codec, XCodec2).
> 2. **Audio Samples:** We strongly encourage listening to the side-by-side samples on our [anonymous demo website](https://sgvqvae.github.io/sgvqvae-demo/), which clearly demonstrate audible reductions in quantization artifacts (e.g., less metallic noise, better high-frequency reconstruction).
>
> Regarding the downstream TTS WER (28–35%): This reflects the difficulty of scratch-training a baseline autoregressive LLM for zero-shot TTS continuation without speaker adaptation. Our goal was a strictly controlled comparison to isolate the codec's impact, not to achieve SOTA TTS. In this setting, the relative improvements of SG over existing approaches are consistent and meaningful.
>
> > **5. Regarding Multi-Codebook VQ**
>
> The drive toward high-quality single codebooks is motivated by the shift toward LLM-based audio generation. Standard LLMs are *natively designed for **single-stream, flat token sequences**. Adapting their **auto-regressive framework** to handle the hierarchical, multi-stream nature of RVQ requires complex workarounds (e.g., MTP modules, delay patterns) that drastically increase sequence length or architectural complexity. Therefore, pushing the limits of a single codebook is a critical research direction for seamless LLM integration.
>
> Furthermore, self-guidance is **not mutually exclusive** with multi-codebook strategies. As shown in Appendix A.4 (Table 7), applying self-guidance to a multi-codebook architecture (Residual FSQ) also yields consistent performance gains. We will clarify this motivation and the relationship with product quantization in the revision.
>
> ---
>
> Thank you for your feedback. We hope these clarifications and updates address your concerns and strengthen the paper.

---

> > ### Author Rebuttal · Reviewer_SQup · 2026-04-03
> >
> > I thank the authors for the detailed response. Most of my concerns have been addressed, however I still have my reservations regarding the perceptual quality evaluation.
> >
> > While objective proxies like UTMOS are used, I am not convinced they can replace a proper subjective evaluation — they are, after all, only proxies.
> >
> > Regarding the audio samples, I think 3 cherry-picked examples are not enough to validate overall performance. It is also not directly clear whether the observed “problems” are truly due to quantization errors. Since the models are trained separately, these differences might simply come from training dynamics or randomness.
> >
> > It would also be better to also show some failure cases of the SG model and compare them with the model without SG in the demo samples.
> >
> > Additionally, one possible evaluation could be to take the SG-trained model and decode using both quantized and pre-quantization latents, and compare the outputs. Even if SG improves robustness, it cannot completely eliminate quantization error, so such an experiment could give more insight into how much robustness is actually achieved in practice.

---

> > > ### Author Response · Authors · 2026-04-08
> > >
> > > Thank you for acknowledging that most of your original concerns have been addressed.
> > >
> > > We conducted the requested subjective evaluation and ablation studies, as well as appending failure case analysis, in order to address your remaining reservations regarding perceptual quality and robustness.
> > >
> > > > **6. Regarding the subjective evaluation**
> > >
> > > To address concerns regarding objective proxies, we conducted a human A/B preference test to verify the actual perceptual quality improvements of self-guidance.
> > >
> > > **A/B preference test setting:** We randomly sampled 30 clips from LibriSpeech test-clean. Listeners indicated their preference regarding reconstruction fidelity compared to the ground truth (GT) reference. Audios from the proposed and baseline models were anonymized and shuffled.
> > >
> > > **Results:** Based on 38 valid evaluations, the results are as follows:
> > >
> > > | | with SG | without SG | No preference |
> > > |-|-|-|-|
> > > | percentage | **38.684%** | 15.351% | 45.965% |
> > >
> > > As shown, self-guidance significantly outperforms the baseline (38.68% vs. 15.35%, a **>2x preference ratio**), confirming that objective metric improvements translate to consistent, significant perceptual enhancements. We will include these details in the final version.
> > >
> > > > **7. Regarding the audio sample cases and failure cases**
> > >
> > > We would like to clarify that the 3 audio samples provided in the original submission were selected as **representative examples**, which illustrate **how** the consistent _quantitative improvements_ translate into actual _perceptual differences_ (e.g., reduced abnormal pitch spikes).
> > >
> > > The _overall performance enhancement_ of the proposed approach, on the other hand, is mainly validated via **massive evaluation** across the following test datasets and the **A/B preference test**:
> > >
> > > 1. 2620 samples from the LibriSpeech test-clean (reported in paper);
> > > 2. 2940 noisy samples from LibriSpeech test-other (in reply to _Reviewer 17Sm (Q3)_);
> > > 3. 2020 Mandarin Chinese samples from SeedTTS Eval ZH test set (in reply to _Reviewer 17Sm (Q3)_).
> > >
> > > Regarding your request for failure cases, we acknowledge your insight that _performance on specific data points can be driven by training dynamics_. Although self-guidance substantially reduces the overall incidence of reconstruction artifacts, it does **not entirely eliminate them**, and achieving perfect reconstruction in all scenarios remains an ongoing challenge for neural codecs.
> > >
> > > Thus, we checked the A/B preference test results for the audio samples where the proposed approach received obvious disfavor.
> > > As a result, **one out of the total 30 samples** is identified: the utterance `61-70968-0004`.
> > > In this case, the reconstructed audio from the proposed approach presents a depressed pitch in the starting word _"also"_ of the utterance.
> > > This issue is similar to the _pitch spike_ issue in the existing sample `4446-2271-0012`, where the pitch contour of the reconstruction audio deviates from the ground truth.
> > >
> > > **We have appended this failure case to our anonymous [demo website](https://sgvqvae.github.io/sgvqvae-demo/)**, so you can listen to the comparison directly.
> > > Furthermore, we will incorporate this case study and the corresponding discussion into the revised paper's appendix to provide a balanced, comprehensive view of the model's capabilities.
> > >
> > >
> > > > **8. Regarding the practical robustness of self-guidance**
> > >
> > > Thank you for suggesting the experiment decoding from both quantized ($z_q$) and pre-quantized ($z_e$) latents. We evaluated the models on LibriSpeech test-clean:
> > >
> > > |With SG|decoder input|WER|SIM|STOI|PESQ-WB|PESQ-NB|MCD|UTMOS|
> > > | --- | --- | --- | --- | --- | --- | --- | --- | --- |
> > > | Y | $z_e$ (pre-quantize) | $\mathbf{3.06}$ | $\underline{0.7919}$ | $\mathbf{0.9186}$ | $\underline{2.33}$ | $\underline{2.94}$ | $\mathbf{3.34}$ | $\underline{4.07}$ |
> > > | Y | $z_q$ (quantized) | $\underline{3.15}$ | $\mathbf{0.8034}$ | $\underline{0.9147}$ | $\mathbf{2.39}$ | $\mathbf{2.98}$ | $\underline{3.41}$ | $\mathbf{4.10}$ |
> > > | - | $z_e$ (pre-quantize) | $\underline{3.15}$ | $0.7857$ | $0.9135$ | $2.21$ | $2.82$ | $3.58$ | $4.05$ |
> > >
> > > The results indicate that:
> > >
> > > 1. SG enables the decoder to **match or surpass** the reconstruction quality of the pre-quantized latent ($z_e$) when using the quantized latent ($z_q$) on PESQ, SIM, and UTMOS. Interestingly, $z_q$ slightly outperforms $z_e$ on some metrics. We hypothesize this occurs because the decoder is optimized to reconstruct from the discrete codebook space during forward passes, and SG successfully aligns this space with high-quality outputs.
> > > 2. Despite minor remaining gaps on ASR-WER, STOI, and MCD, the quantized latent with SG **still significantly outperforms the baseline's upper limit across all metrics** (reconstructing from the pre-quantized latent without SG).
> > >
> > > We will include the results and analysis in the revised paper.
> > >
> > > ---
> > >
> > > We hope these updates resolve your concerns and kindly ask for a positive reconsideration of our work.

---

### Official Review · Reviewer_xJ8V · 2026-03-08

**Soundness:** 3
**Presentation:** 3
**Significance:** 3
**Originality:** 3
**Overall Recommendation:** 4
**Confidence:** 4

**Summary:**

This paper proposes using the latent variables before quantization as the input to the decoder and employing reconstruction error as an auxiliary metric. Experimental results show that this method improves XCodec2 and enhances the efficiency of LLM-based applications.

**Compliance With Llm Reviewing Policy:**

Affirmed.

**Final Justification:**

I increased my score.

**Key Questions For Authors:**

Are the five contributions mentioned in the paper sufficiently independent?

What criteria were used for selecting the audio visualization samples? Which samples are more appropriate to present to readers?

**Limitations:**

see weakness

**Strengths And Weaknesses:**

Strengths:

Intuitive and efficient method: The proposed approach is simple and easy to implement.

Sufficient validation: Ablation studies and effectiveness analyses are provided, demonstrating improvements over XCodec2.

Rich supplementary materials: The paper includes extensive spectrogram visualizations and listening examples, which help illustrate the source of the improvements.

Weaknesses:

Insufficient justification for the “Manifold” claim: Although the title and contributions emphasize manifold alignment, the term appears infrequently in the paper and lacks direct evidence supporting the claim of “manifold alignment.” Currently, the evidence—mainly the mean shift in Fig. 3(b) and the MSE metric—seems somewhat insufficient. It would be beneficial to provide additional evidence, such as local structure preservation or correspondence accuracy.

Limited explanation of experimental phenomena: Some experimental observations, such as the increased density of large-value distributions and the degradation of the baseline when the codebook size increases, lack clear explanations. In addition, the reasons why SG does not achieve the best performance on certain metrics should be further analyzed.

Theoretical depth could be strengthened: If the work aims to highlight manifold properties, it would be helpful to include theoretical assumptions or propositions to justify potential improvements in generalization bounds. Furthermore, it should be clarified whether the latent variables act as supplementary information or as a strong constraint during decoder training.

Room for improvement in presentation: The manuscript contains unnecessary whitespace, repeated links, and excessive use of footnotes. The main text could accommodate more visualizations rather than relying heavily on the appendix, which would improve the overall completeness.

Conclusion section could be improved: The conclusion should be more concise and avoid repeating content from the abstract and introduction. It would also benefit from discussing limitations and potential future directions so that readers can better assess the paper’s reference value.

Suggestions on formatting and details: It is recommended to unify the format of section summaries, improve figure layout and space utilization (e.g., Fig. 1), standardize symbol usage (e.g., × vs. x), ensure consistency between figures and citations, and unify the formatting of references.

---

> ### Author Rebuttal · Authors · 2026-03-31
>
> Thank you very much for your detailed and insightful review. We address each of your concerns below.
>
> > **1. Regarding justification for manifold alignment**
>
> In addition to MSE metrics, we conducted two new analyses providing stronger evidence for manifold alignment, which we will include in the camera-ready version.
>
> 1. **Two complementary manifold alignment metrics** on hidden features. Self-guidance significantly improves both local neighborhood structure (kNN Jaccard) and global geometric alignment (Procrustes), supporting the manifold alignment claim beyond simple MSE reduction.
>
> - **kNN Jaccard similarity (higher is better):** Measures if the same points are neighbors across the two manifolds (teacher vs. student).
> - **Procrustes residuals (lower is better):** Measures overall geometric similarity after optimal rotation.
>
> |with SG|kNN Jaccard |Procrustes residuals|
> |-|-|-|
> |Y|**0.307**|**0.171**|
> |-|0.276|0.265|
>
> 2. **A block-wise analysis of manifold alignment** across all 12 transformer blocks in the decoder. The results show that self-guidance substantially improves alignment throughout the decoder.
>
> - **Spearman correlation (higher is better)** of pairwise distances in hidden space between teacher and student.
> - **linear CKA (Centered Kernel Alignment, higher is better)**.
>
> |Block|with SG|Spearman|linear CKA|
> |-|-|-|-|
> |0|Y|**0.920**|**0.935**|
> |0|-|0.747|0.888|
> |5|Y|**0.929**|**0.977**|
> |5|-|0.737|0.863|
> |6|Y|**0.927**|**0.977**|
> |6|-|0.735|0.874|
> |7|Y|**0.923**|**0.979**|
> |7|-|0.807|0.910|
> |8|Y|**0.925**|**0.981**|
> |8|-|0.828|0.931|
> |9|Y|**0.927**|**0.980**|
> |9|-|0.830|0.941|
> |10|Y|**0.934**|**0.979**|
> |10|-|0.864|0.954|
> |11|Y|**0.946**|**0.978**|
> |11|-|0.933|0.965|
>
> > **2. Regarding clarification on experimental phenomena**
>
> We appreciate your attention to these details. We have re-examined the results to infer the intended observations:
>
> 1. **Figure 3(b):** The increased density in the large-value tail is observed **in the baseline (blue)**. Our method (orange) successfully suppresses this effect, keeping the distribution concentrated.
> 2. **Table 3 baseline UTMOS:** The score fluctuates (4.08 → 3.98 → 4.06 for 8k → 16k → 65k). This is a known phenomenon where larger codebooks introduce unnatural artifacts (e.g., oversmoothing) that UTMOS penalizes, even as fidelity metrics improve. SG stabilizes this trend.
> 3. **Table 3 WER (16384):** A slight WER increase with SG is observed only at codebook size 16,384, which may reflect a critical trade-off between semantic and acoustic information at this bottleneck. Notably, SG still yields substantial improvements across all other metrics (PESQ, STOI, SIM, UTMOS).
>
> We will add these explanations to the camera-ready version.
>
> > **3. Regarding theoretical depth**
>
> We will add the following clarifications to the paper:
>
> 1. **Role of latents:** The continuous pre-quantized latents act as a soft constraint. Since the stop-gradient prevents the encoder from being affected, the decoder is regularized to learn a mapping from the quantized space to the same output as the continuous space.
> 2. **Theoretical intuition:** We will add a brief proposition: *Assuming the decoder is Lipschitz continuous, minimizing $|h_e - h_q|^2$ bounds the reconstruction error difference between the two paths: $|\hat{x}_e-\hat{x}_q| \leq L\cdot|h_e - h_q|$, where $L$ is the Lipschitz constant of the decoder's output layer. Thus, aligning intermediate features indirectly bounds the output discrepancy.*
>
> > **4. Regarding presentation and writing improvements**
>
> Thank you for your suggestions. In the camera-ready version, we will:
> 1. Condense the main text: summarize contributions, reduce redundancy in the conclusion, optimize Figure 1 layout.
> 2. Move key tables and visualizations (e.g., t-SNE, manifold alignment) from the appendix to the main text.
> 3. Remove unnecessary whitespace, symbols, and unify citation formatting.
>
> > **5. Regarding the conclusion**
>
> We will revise the conclusion to be more concise and outline future directions: (i) extension to music, general audio, and images; (ii) theoretical bounds.
>
> > **Q1. Regarding 5 contributions**
>
> Thank you for your question. We will condense the contributions into 3 distinct points:
> 1. Proposing the self-guidance mechanism and providing statistical evidence of manifold alignment.
> 2. Applying SG to XCodec2 to achieve SOTA, and demonstrating generalization across configurations.
> 3. Showing that smaller codebooks benefit downstream LLMs.
>
> > **Q2. Regarding audio samples**
>
> The samples in Figures 4–6 were selected for:
> 1. **Representativeness**: They exhibit common quantization artifacts (smeared harmonics, pitch spikes, oversmoothing).
> 2. **Clarity**: Artifacts are visually and audibly distinct.
> 3. **Diversity**: They cover different speakers and utterances in the test dataset.
>
> We will add these criteria to the revised paper.
>
> ---
>
> Thank you for your feedback. We hope these clarifications address your concerns and strengthen the paper.

---

> > ### Author Rebuttal · Reviewer_xJ8V · 2026-04-01
> >
> > Thank you for the authors’ response; my concerns have been largely addressed, and I will increase my score.

---

### Official Review · Reviewer_tVs1 · 2026-03-10

**Soundness:** 3
**Presentation:** 3
**Significance:** 3
**Originality:** 3
**Overall Recommendation:** 5
**Confidence:** 4

**Summary:**

The authors propose a self-guidance training framework for VQ-VAEs in the context of audio tokenizers for LLMs that is meant to mitigate the performance drop that comes with bottleneck quantization. This self-guidance framework operates by comparing hidden features generated by the decoder from a quantized latent representation to features generated by an unquantized latent representation. The method improves reconstruction performance metrics across two architectures, XCodec2 and BigCodec, while not leading to significantly increased training cost.

**Compliance With Llm Reviewing Policy:**

Affirmed.

**Final Justification:**

The authors addressed my primary concerns with the breadth of their evaluation, as well as the originality of their proposed method. As such I have increased the rating to an Accept, and increased the Originality score.

**Key Questions For Authors:**

1) Are there other open source VQ-VAE Neural Audio Codecs that can benefit from this training regime?
2) Does this training approach hold for different internal frame rates?
3) Were other layers of the decoder considered for hidden feature alignment loss?
4) What are the effects of using a different type of gradient propagation, i.e. the 50% mix proposed in TAAE?

**Limitations:**

The authors are limited by testing their method on only two architectures at a single frame rate.

**Strengths And Weaknesses:**

Soundness:

The self-guidance experiments in the paper are technically sound. The core method is well-motivated, and the main hypothesis, that including this additional feature loss when training VQ-VAEs, is supported by the experiments. The provided audio samples highlight perceptual improvements self-guidance can make in a head-to-head comparison. The ablation presented in 5.4 is also quite good and supports their claims, though crucially does not vary framerate nor investigate other layers of the decoder for alignment loss.

Claims that “a large flat audio token vocabulary is harmful to the downstream LLMs” feel rushed and technically unsound given the small scale of their evaluation. This seems to be an important hypothesis of theirs and warrants a fuller treatment than presented here.

In general, the primary weakness of this paper lies in the limited scope of its evaluation. Specifically, the claim of a “general purpose” method is hampered by evaluating using two architectures at one framerate. Incorporating internal representation feature loss has been shown to improve speech synthesis systems for quite some time, so a more robust evaluation would help delineate how well this approach serves neural audio codecs for speech LLMs. I am willing to revise my assessment here should the authors provide evidence that training other VQ-VAEs from scratch is infeasible given open-source restrictions.

Others have attempted to address the bottleneck of VQ-VAE quantization with different strategies, for example the 50% mix proposed in “Scaling Transformers for Low-Bitrate High-Quality Speech Coding” (Parker 2024)




Presentation:

The paper is well written and well structured with some room for improvement in terms of clarity.
1) The authors should clarify if Table 1 is a reproduction of a table from another paper, or is an experiment they ran themselves.
2) After equation (1), the authors state “This error represents the information loss incurred during discretization.” This should be revised to something closer to how the error represents lost precision in the representation, as the entire thrust of this paper is that information can be recovered, despite the quantization, using self-guidance
3) The authors should clarify their statement of “differences only in seconds” in the Training Cost section with a proper table with measurements, as this is another important claim the paper is making




Significance:

The paper addresses a relevant problem in the field of VQ-VAEs. Should this approach hold with other VQ-VAE architectures, it would represent a drop-in improvement to any training regime.




Originality:

This is an original approach to the best of my knowledge in its application to VQ-VAE codebooks for speech LLMs. Incorporating hidden feature losses to improve speech generation have been published for some time now, at least since MelGAN (Kumar 2019).

---

> ### Author Rebuttal · Authors · 2026-03-31
>
> Thank you very much for your detailed and constructive review. We address each of your concerns below.
>
> ---
>
> > **1. Regarding evaluation scope (Weakness and Q1, Q2)**
>
> Thank you for raising this point. Relative to a general-purpose claim, we clarify that our study already stress-tests **public (open-source) neural codecs** along **several distinct inductive axes**, not a single stack:
>
> 1. **Decoder family**: **XCodec2** (Transformer decoder; main results) and **BigCodec** (CNN/RNN-style decoder; Appendix A.5, Table 9)—i.e., open, recent architectures that reflect the dominant neural-decoder paradigms.
>
> 2. **Temporal resolution**: **50 Hz** (XCodec2) vs. **40 Hz** (BigCodec in our setup), varying internal frame rate.
>
> 3. **Codebook width**: **8,192 / 16,384 / 65,536** (Table 3).
>
> 4. **Quantization modeling**: **FSQ** plus **SimVQ** (projector-based; Appendix A.3) and **Residual FSQ** (multi-codebook; Appendix A.4).
>
> We are aware that adding further open repositories would sharpen external validity; in this work, we prioritized breadth along **architecture, frame rate, codebook scale, and quantizer design**, each requiring full retraining and strict recipe alignment. We will state this scope explicitly in the camera-ready version and flag systematic sweeps over additional codecs as important future work.
>
> > **2. Regarding the originality of the self-guidance**
>
> Thank you for recognizing the novelty of applying this idea to VQ-VAE speech codecs. We clarify how self-guidance relates to MelGAN-style feature matching.
>
> 1. **Feature matching (e.g., MelGAN)** is an adversarial auxiliary loss: it compares intermediate **discriminator** activations on real versus generated waveforms and steers the generator during GAN training.
> 2. **Self-guidance** is confined to the **codec decoder**: the continuous pre-quantized representation acts as a teacher for the quantized branch, directly targeting quantization error rather than discriminator realism.
>
> These mechanisms are **distinct and complementary**: MelGAN-style feature matching is already denoted as **part of** the $\mathcal{L}_{\mathrm{adv}}$ term of our training objective in Equation 3, whereas self-guidance is an additional, **decoder-internal** term that targets quantization robustness.
>
> > **3. Regarding the claim that "a large flat audio token vocabulary is harmful"**
>
> Thank you for your suggestion. We will revise the text as follows to properly scope our finding:
>
> _"Holding LLM capacity and training budget fixed, smaller speech codebooks yield better downstream TTS in our experiments. We interpret this as a harder LM objective when acoustic tokens form a very large flat vocabulary; whether the trend persists at larger scale and across tokenizers is important future work."_
>
> > **4. Regarding the source of results of Table 1**
>
> Thank you for asking. The results for BigCodec and XCodec2 are from our own reproduction experiments. The rest are from the relevant papers. We will add a clear footnote in the camera-ready version to avoid ambiguity.
>
> > **5. Regarding the wording of Equation 1**
>
> Thank you for your insightful distinction. We will revise the sentence:
>
> *"This error summarizes the deviation of the quantized representation from the original continuous one, providing a measure of the precision loss of the decoder input caused by discretization."*
>
> > **6. Regarding the precise measurement training cost**
>
> In our experiments on 8 NVIDIA 4090 GPUs, the training cost on LibriSpeech for XCodec2 was:
>
> |with SG|average training time per epoch|
> |-|-|
> |-|25783.8 secs|
> |Y| **25668.0 secs**|
>
> The observed difference (<0.5%) is minimal and within regular system fluctuations. We will include this table in the camera-ready version.
>
> > **Q3. Regarding using other layers for SG loss**
>
> Due to the content length limit, we kindly refer you to our response to _Reviewer 17Sm (Q1)_ for a full metric table and alignment figures. In brief:
> 1. Applying SG to intermediate layers (e.g., 6th block) **degrades** performance, likely because it dominates gradients over the reconstruction objective.
> 2. Applying SG **only at the final output** naturally improves manifold alignment across all earlier decoder blocks.
>
> > **Q4. Regarding the effects of 50% mix proposed in TAAE**
>
> Thank you for highlighting TAAE. Its 50% mix strategy is an elegant approach to softening scalar quantization, enabling **flexible post-training adjustment of scalar quantization levels** (analogous to layer dropout in RVQ).
>
> In contrast, self-guidance incorporates an additional teacher branch only during training **to enhance decoder robustness to quantization error**, incurring no gradient propagation. It is agnostic to the underlying VQ mechanism and gradient propagation of the student branch, **including the TAAE**.
>
> We will prominently include this discussion in the camera-ready version, citing TAAE.
>
> ---
>
> Thank you again for your constructive feedback, which will greatly strengthen the final paper.

---

> > ### Author Rebuttal · Reviewer_tVs1 · 2026-04-03
> >
> > I appreciate the authors' comprehensive and clarifying responses. I have upgraded my rating accordingly.

---

### Official Review · Reviewer_17Sm · 2026-03-17

**Soundness:** 3
**Presentation:** 4
**Significance:** 4
**Originality:** 3
**Overall Recommendation:** 5
**Confidence:** 5

**Summary:**

Self-Guidance (SG) is an efficient training framework that enhances the reconstruction fidelity of neural speech codecs by aligning the decoder's internal feature representations. Recognizing that VQ-VAE decoders inherently process continuous latent embeddings more effectively than their quantized counterparts, the authors introduce a secondary training path where continuous embeddings serve as a "teacher" for the quantized "student" path within the same model. By minimizing a manifold alignment loss ($\mathcal{L}_{guide}$) between the intermediate features of these two paths, the decoder learns to recover information lost during quantization without requiring additional parameters or changing the inference-time architecture. This approach boosts audio quality and enables a four-fold reduction in codebook size, effectively simplifying the token space for downstream speech language models while maintaining state-of-the-art performance.

**Compliance With Llm Reviewing Policy:**

Affirmed.

**Final Justification:**

Increasing my score to reflect the strengthened evidence and the practical significance of this work.

**Key Questions For Authors:**

- Ablation on Alignment Layer: The authors chose the output of the final Transformer block for $\mathcal{L}_{guide}$. Did you experiment with aligning earlier layers or multiple layers simultaneously? How does the choice of the alignment layer impact the trade-off between semantic preservation and acoustic fidelity?

- Since SG allows for a 4× smaller codebook, have you analyzed the codebook utilization (perplexity) to see if SG effectively "packs" more information into fewer entries or simply makes the decoder better at interpolating between them?

- Cross-Domain Generalization: The results are on LibriSpeech (clean English). Have you tested how SG handles noisy environments or different languages, where the quantization error might be more "unpredictable", making it harder for the alignment loss to stabilize?

**Limitations:**

I like that the innovation is not in the alignment or distillation but in the architecture-agnostic, single-stage application of these concepts to solve the specific manifold mismatch in VQ-VAE decoders. However, a primary limitation is the potential lack of generalizability across varying inductive biases; the method is validated on Transformer-based decoders, yet it remains unproven whether this feature-mapping loss effectively translates to Convolutional Neural Network (CNN) architectures without introducing "checkerboard" artifacts or conflicting with local receptive fields. Furthermore, we need to know whether the alignment improves perceived naturalness or simply forces a "feature collapse" where phonetic nuances are smoothed over to minimize mathematical distance.

**Strengths And Weaknesses:**

## Strengths:

- The proposed method is technically sound and addresses the "quantization artifact" problem through a well-justified manifold alignment perspective. The inclusion of a stop-gradient on the continuous path ensures that the encoder is not adversely affected, focusing the improvement on the decoder's robustness.

- The paper is exceptionally clear and well-structured. Figure 1 provides an intuitive comparison between vanilla VQ-VAE and the SG-enhanced version. I also like the GitHub demo added by the authors, though I couldn't hear the audible difference between the baselines and the proposed method.

- The work is highly significant for the speech LLM community. By enabling a 4× reduction in codebook size while maintaining quality, it drastically simplifies the vocabulary space for downstream autoregressive modeling, reducing computational costs and modeling complexity.


## Weaknesses:

- Other SOTA codec methods have used CNNs. It would be great if the authors could provide ablations on other architectures. Without this, the claim of "universal applicability" is overstated.

- There is a tiny risk that this alignment could lead to "feature collapse", where the decoder might learn to output something constant to minimize MSE. Can authors provide t-SNE/UMAP visualizations of the latent space to demonstrate that the alignment is occurring?

- Dual path forward pass will add some training overhead. It will be nice to know the additional resources (GPU/time) required for the training.

- The authors do not provide an extensive sensitivity analysis for this hyperparameter. It will be helpful to understand whether the method shows stable gains across a wide range of loss-scaling hyperparameters.

## Minor issues:

- Figure 3 caption: "alignemnt" -> "alignment"
- Section 5.1, Implementation details: "offical" -> "official"
- Section 3.2: "know as" -> "known as"

---

> ### Author Rebuttal · Authors · 2026-03-31
>
> Thank you for your thorough review and for recognizing our work's clarity and significance. We address your concerns below.
>
> ---
>
> > **1. Regarding generalization across varying inductive bias**
>
> We have validated our method's "universal applicability" across diverse architectures, where self-guidance (SG) shows consistent gains. To make this clearer:
>
> 1. **CNN-based codec (BigCodec)**: Table 9 (p. 15), Appendix A.5.
> 2. **projector-based VQ (SimVQ)**: Table 8 (p. 15), Appendix A.3.
> 3. **multi-codebook VQ (Residual FSQ)**: Table 7 (p. 14), Appendix A.4.
>
> We briefly noted these in the experiment section of the main paper; full details are in the Appendix. In the camera-ready version, we will reorganize the Appendix to improve readability.
>
> > **2. Regarding the risk of feature collapse**
>
> Thank you for raising this concern. We performed t-SNE on decoder hidden features from the final hidden state ([figure link](https://github.com/sgvqvae/sgvqvae-demo/blob/sgvqvae-add-pics/pics/tsne.png)):
>
> 1. **With SG**: Hidden features from both the continuous teacher (circles) and quantized student (triangles) cluster by token ID (color, 50 most common tokens), indicating that SG preserves discriminative, token-specific information and does not induce feature collapse.
> 2. **Without SG (complementary context)**: Features from the two branches separate into distinct halves of the latent space (red dashed line), showing baseline manifold misalignment. This is not a collapse, but it visualizes the dual-path inconsistency SG is designed to mitigate (consistent with the paper’s motivation).
>
> We will include this t-SNE in the camera-ready version. We appreciate this valuable suggestion.
>
> > **3. Regarding the additional training overhead**
>
> Due to content length limit, we kindly refer you to our response to _Reviewer tVs1 (point 6)_, where self-guidance adds negligible training cost (<0.5%).
>
> > **4. Regarding the sensitivity analysis for the guidance loss weight**
>
> λ_guide is validated in Appendix A.2, Table 6.
>
> A brief conclusion:
> 1. Optimal range is 5–10, with robust gains;
> 2. λ_guide=1 yields minimal effect;
> 3. λ_guide≥15 degrades performance.
>
>
> > **Q1. Regarding the ablation on the alignment layer**
>
> Thank you for this insightful question. We conducted two additional analyses:
>
> 1. We evaluated applying the guidance loss to both the final and 6th transformer blocks (of 12), and observed that performance declined across all metrics in this setting. As the reconstruction objective targets **only the final output**, we hypothesize that SG loss on intermediate layers may disproportionately influence gradients and impair reconstruction.
>
> |SG Layer|PESQ-WB|PESQ-NB|STOI|MCD|WER|SIM|UTMOS|
> |-|-|-|-|-|-|-|-|
> |Final (default)|**1.99**|**2.53**|**0.89**|**3.93**|**4.00**|**0.706**|**3.57**|
> |Final + 6th block|1.96|2.50|0.88|3.96|4.01|0.703|3.53|
>
> 2. To confirm that **aligning only the final output already benefits earlier layers**, we computed block-wise linear CKA to measure manifold alignment between teacher and student branches across all 12 decoder blocks. As shown in [this figure link](https://github.com/sgvqvae/sgvqvae-demo/blob/sgvqvae-add-pics/pics/blockwise_align.png), SG at the final output leads to substantial alignment improvements throughout the network.
>
> We will include these analyses in the camera-ready version.
>
> > **Q2. Regarding the codebook utilization analysis**
>
> We computed the codebook utilization rate:
>
> |Codebook size|SG|Rate(%)|
> |-|-|-|
> |16384|-|97.12|
> |16384|Y|**97.82**|
> |65536|-|96.60|
> |65536|Y|**97.11**|
>
> While SG yields a modest improvement in utilization, the difference is not dramatic. The primary mechanism is making the decoder more robust to quantization error, as shown by our manifold alignment analysis.
>
> > **Q3. Regarding cross-domain generalization**
>
> We evaluated the model on two generalization scenarios as follows. The results show that SG consistently improves performance across all metrics and codebook sizes, demonstrating strong cross-domain robustness.
>
> - **LibriSpeech test-other: noisy speech**
>
> |Codebook size|SG|PESQ-WB|PESQ-NB|STOI|MCD|WER|SIM|UTMOS|
> |-|-|-|-|-|-|-|-|-|
> |16384|-|2.01|2.56|0.875|3.90|10.20|0.736|3.55|
> |16384|Y|**2.11**|**2.66**|**0.880**|**3.80**|**9.97**|**0.752**|**3.62**|
> |65536|-|2.12|2.69|0.884|3.79|9.25|0.769|3.61|
> |65536|Y|**2.20**|**2.76**|**0.889**|**3.64**|**8.50**|**0.783**|**3.65**|
>
> - **Seed-TTS Eval ZH: Mandarin Chinese**
>
> |Codebook size|SG|PESQ-WB|PESQ-NB|STOI|MCD|WER|SIM|UTMOS|
> |-|-|-|-|-|-|-|-|-|
> |16384|-|1.89|2.45|0.87|4.38|3.41|0.59|2.91|
> |16384|Y|**1.93**|**2.60**|**0.88**|**4.33**|**3.37**|**0.61**|**2.98**|
> |65536|-|2.01|2.58|0.88|4.22|2.28|0.61|2.98|
> |65536|Y|**2.08**|**2.67**|**0.89**|**4.08**|**2.18**|**0.62**|**3.05**|
>
> ---
>
> We appreciate your valuable feedback, including the writing issues; we will correct all typos in the camera-ready version.
>
> We hope these clarifications and experiments address your concerns and further strengthen the paper.

---

> > ### Author Rebuttal · Reviewer_17Sm · 2026-04-05
> >
> > Thank you for the rebuttal. I appreciate the additional experiments on CNN architectures and the cross-domain evaluations on noisy speech and Mandarin, which successfully address my concerns regarding universal applicability.
> > The t-SNE visualization and the block-wise CKA analysis provide strong evidence that manifold alignment is preserved without feature collapse. Furthermore, the clarification that training overhead is $<0.5\%$ mitigates my concerns regarding computational cost.
> > I have increased my score to reflect the strengthened evidence and the practical significance of this work.

---

### Decision · Program_Chairs · 2026-04-30

**Decision:**

Accept (regular)

**Comment:**

The paper tackles the problem of improving the robustness of VQ‑VAE–based neural speech codecs to quantization error, which is a key bottleneck for high‑fidelity speech tokenization and for using these codecs as front-ends to speech LLMs and TTS systems.

Most reviewers summarize that the work proposes a simple self‑guidance training mechanism backed by strong empirical evidence showing consistent improvements across several standard objective metrics, spanning different codec architectures/quantizers, codebook sizes, and datasets. The 4× reduction in codebook size without loss of reconstruction quality is particularly noteworthy, given the negligible training overhead and the absence of any increase in inference-time cost.

Based on the consensus of the reviewers, I recommend acceptance of this work as a solid contribution. I would request that the authors take into account the following reviewer feedback for the camera-ready version:

- Soften and better scope the “manifold” / “universal codec enhancer” language so that claims more closely match the presented evidence.
- Move key manifold-alignment analyses, t‑SNE plots, and subjective evaluation protocol/results into the main paper rather than relegating them to the appendix or external material.
- Clarify limitations: current validation is confined to neural speech codecs (no image or non-audio domains), and downstream LLM/TTS results, while directionally positive, are not yet exhaustive.
- Improve presentation and layout (whitespace, figure placement, and conclusion), and use the conclusion to clearly discuss limitations and future work.